
# Assessing uncertainties in landslide susceptibility predictions in a changing environment (Styrian Basin, Austria)

Raphael Knevels[1], Helene Petschko[1], Herwig Proske[2], Philip Leopold[3], Aditya N. Mishra[4], Douglas Maraun[4], and Alexander Brenning[1]

[1]Department of Geography, Friedrich Schiller University Jena, Jena, 07743, Germany
[2]Remote Sensing and Geoinformation Department, JOANNEUM RESEARCH Forschungsgesellschaft mbH, Graz, 8010, Austria
[3]Centre for Low-Emission Transport, AIT Austrian Institute of Technology GmbH, Vienna, 1210, Austria
[4]Wegener Centre for Climate and Global Change, Regional Climate Research Group, Karl-Franzens-University Graz, Graz, 8010, Austria

**Correspondence:** Raphael Knevels (raphael.knevels@uni-jena.de)

**Abstract.** The assessment of uncertainties in landslide susceptibility modelling in a changing environment is an important, yet often neglected task. In an Austrian case study, we investigated the uncertainty cascade in storylines of landslide susceptibility emerging from climate change and parametric landslide model uncertainty. In June 2009, extreme events of heavy thunderstorms occurred in the Styrian basin, triggering thousands of landslides. Using a storyline approach, we discovered a

generally lower landslide susceptibility for pre-industrial climate, while for future climate (2071–2100) a potential increase of 35 % in highly susceptible areas (storyline of much heavier rain) may be compensated by much drier soils ($-45$ % areas highly susceptible to landsliding). However, the estimated uncertainties in predictions were generally high. While uncertainties related to within-event internal climate model variability were substantially lower than parametric uncertainties of the landslide susceptibility model (ratio of around 0.25), parametric uncertainties were of the same order as the climate scenario uncertainty

for the higher warming levels ($+3$ K and $+4$ K). We suggest that in future uncertainty assessments, an improved availability of event-based landslide inventories and high-resolution soil and precipitation data will help to reduce parametric uncertainties of landslide susceptibility models used to assess the impacts of climate change on landslide hazard and risk.

## 1  Introduction

Climate and land use/land cover (LULC) are changing worldwide, altering the risk of landslide occurrences. During the period

from 1998 to 2017, landslides affected 4.8 million people causing more than 18,000 deaths and over 5.28 billion US$ total damages (Wallemacq et al., 2018). In the future, these landslide-associated casualties are likely to increase globally (Haque et al., 2019; Gariano and Guzzetti, 2021), and such an increase can already be observed in some regions (Schlögel et al., 2020). In the Alpine region and in Austria in particular, landslides are among the main natural hazards frequently causing damage to houses and infrastructure as well as casualties. Here they are conditioned by local geomorphology, geology and LULC,

with long-lasting heavy rainfall and rapid snowmelt as the main natural triggers (Schweigl and Hervás, 2009). In south-eastern



Austria extreme precipitation is projected to increase by up to 14 percent for each Kelvin of warming ($\% \, K^{-1}$) as a consequence of climate change, which was found to affect the risk of landslide occurrences (Olefs et al., 2021; Maraun et al., 2022).

In June 2009 and September 2014, weather phenomena developed through a cut-off low brought extreme rainfall into the Styrian Basin, Austria (e.g., over 100 mm in 24 h in 2009). In both events in total more than three thousand landslides were
triggered, causing significant damage to human infrastructure (e.g., about 13.4 million € in 2009; Hornich and Adelwöhrer 2010; ZAMG 2014). The combined effect of premoisturing over the preceding winter and spring, and the actual triggering rainfall made these extreme events into compound events (Zscheischler et al., 2020; Maraun et al., 2022). In addition to the high amount of rain, local experts identified human activities as a conditioning factor (steep or unsecured embankments, artificial slope surcharge, impervious paved surfaces; Hornich and Adelwöhrer 2010). Statistical investigations of these landslide events
confirmed the important role of meteorological (rainfall intensity and five-day rainfall) and LULC (forest types) predictors (Knevels et al., 2020). Focusing on the rainfall event in 2009, Maraun et al. (2022) analysed the effect of projected future climate (2070–2100) and LULC changes on landslide occurrences using a storyline approach for the most-affected Feldbach region. While cut-off lows would become slightly less frequent, the area threatened by landslides, given such an event, would increase by 45 % in a 4 K global warming scenario (Maraun et al., 2022). However, a comprehensive assessment of uncertainties
inherent in landslide susceptibility predictions has yet to be conducted, which is the objective of our study.

Uncertainty assessments are essential for the development of business strategies and policy interventions, they increase transparency and confidence in scientific analyses, and they are considered a 'good modelling practice' (Kirchner et al. 2021 and references therein). Generally, depending on the scientific community, different definitions of uncertainties exist (Walker et al., 2003; Refsgaard et al., 2007). In risk assessment, uncertainty is commonly categorised into aleatory and epistemic un-
certainty (Oberkampf et al., 2004; Roy and Oberkampf, 2011; Rougier et al., 2013). Aleatory uncertainty refers to the natural variation or randomness inherent in the natural hazard process, and is thus irreducible and unavoidable (Roy and Oberkampf, 2011; Rougier, 2013). Epistemic uncertainty arises from missing knowledge, and is reducible by enhancing knowledge on the subject (Roy and Oberkampf, 2011; Rougier and Beven, 2013). Additionally, three sources of epistemic uncertainty can be distinguished (Rougier and Beven, 2013): input uncertainty (e.g., initial or boundary condition uncertainty in climate mod-
elling), parametric uncertainty (e.g., uncertainty in model parameter settings), and structural uncertainty (e.g., uncertainty in model form to represent the system). However, according to Roy and Oberkampf (2011) the boundary between aleatory and epistemic uncertainty is fluid and depends on the question. Furthermore, when different sources of uncertainties propagate through a sequence of calculations, the full range of uncertainties is referred to as the uncertainty cascade (Refsgaard et al., 2016).

In landslide risk management, many authors have emphasised the importance of the quantification of uncertainties, especially considering environmental change (Reichenbach et al., 2018; Gariano and Guzzetti, 2021). However, only a tiny fraction of around 3 % of recent statistical landslide susceptibility analyses actually address this issue (Reichenbach et al. 2018, based on 565 articles published between 1983 and 2016). Focusing on statistically-based landslide susceptibility analysis, input and parameter uncertainties have previously investigated for landslide inventory biases or incompleteness (e.g., Steger et al.,
2016, 2017), the quality, resolution and spatial scale of input data (e.g., Guzzetti et al., 2006; Fressard et al., 2014; Torizin





et al., 2021), and the size and strategy effects of landslide sampling (e.g., Petschko et al., 2014; Hussin et al., 2016). For the assessment of structural uncertainty, model validation procedures such as (spatial) cross-validation are well-established assessment techniques (e.g., Petschko et al., 2014; Torizin et al., 2021). However, only few studies account explicitly for often invisible effects of spatial (and temporal) autocorrelation in landslide model fitting and prediction (Reichenbach et al.,
2018; Lombardo et al., 2020). Spatially varying uncertainties in landslide susceptibility predictions, emerging from parametric uncertainty (or sampling variability) and model error, have been assessed in terms of the standard error (or confidence interval) of the predicted probabilities (Guzzetti et al., 2006; Petschko et al., 2014), or by model ensembles (Kim et al., 2018; Felsberg et al., 2021).

Our objective was the assessment of uncertainties in landslide susceptibility predictions considering both parametric land-
slide model uncertainty and climate change uncertainty (within-event internal model variability and scenario uncertainty), while accounting for different storylines of LULC and climate change in the Styrian basin, Austria. Such a joint consideration of uncertainties in integrated modelling (or interdisciplinary system of linked models) is still an important research gap (Kirchner et al., 2021), and existing studies mostly come from a non-landslide context (e.g., hydrological or crop modelling; Bastola et al. 2011; Holzkämper et al. 2015; Refsgaard et al. 2016). In our analysis, we built upon an existing landslide susceptibility
model (Maraun et al., 2022) that linked different predisposing and triggering factors to the 2009 and 2014 landslide occurrences. Our workflow combines within-event internal climate model variability with probabilistic simulations of parametric uncertainties of the landslide model into an uncertainty cascade in order to obtain a storyline uncertainty assessment. With this approach, we differentiate the uncertainty components in the analysed landslide susceptibility predictions.

## 2 Study Area and Data

### 2.1 Study Area and Extreme Rainfall Events

Landslides in the Styrian Basin, Austria, have received much attention especially recently after the 2009 and 2014 compound events (Hornich and Adelwöhrer, 2010; Knevels et al., 2020; Maraun et al., 2022). Our study area (3831 km$^2$) is characterised by flat lowlands in the East and a hilly topography in the West, ranging from 196 to 1167 metres above the Adriatic Sea (m AA) with a relative relief of 3–550 m km$^{-2}$. The Styrian Prealps border the study area from southwest (Possruck Mountains,
Koralpe) to northwest (Stubalpe Mountains, Graz Mountains). According to Gasser et al. (2009) and Hornich and Adelwöhrer (2010) the Styrian basin has a high predisposition to landslides due to the underlying geology of Neogene sediments (thick Miocene and minor Pliocene sediments) mainly consisting of a heterogeneous mixture and interbedded strata of sands, silts, clays, marl, and gravels (more geological details are given in Gasser et al. 2009).

In June 2009 and September 2014, extreme rainfall events occurred in southeast Styria. From June 22 to 25, 2009, a series
of heavy thunder- and rainstorms brought over 100 mm of precipitation within 24 hours in some places, which corresponds to a 50-year return period (Fig. 1c left; meteorological details in Haiden 2009; Hornich and Adelwöhrer 2010). In the region, around 1700 landslide-related private damage claims were submitted to the state government, surpassing 13.4 million € of cost for reconstruction and emergency response (Hornich and Adelwöhrer, 2010). The northern part of the district of South East

**Figure 1.** Overview of the study area. (a) Location in Austria (upper part) and federal state of Syria (lower part). (b) Occurred landslides during extreme rainfall events. (c) Accumulated rainfall [mm] of the event in 2009 (left) and 2014 (right). Adopted from Knevels et al. (2020).


Styria was particularly severely affected (Fig. 1b), so that for several municipalities the state of emergency had to be declared

(Hornich and Adelwöhrer, 2010). The rainfall event in 2014 was similar, although less severe. In September 2014, several heavy thunderstorms occurred within a three-week period (ZAMG, 2014). From September 12 to 15, between 30 and 100 mm of rain (Fig. 1c right) was brought to the region by an upper-level low (Steiermark, 2014). Again, landslides occurred across the entire Styrian basin, but were clustered with more than 500 landslides south of Leibnitz and north of Gleisdorf (Fig.1b).

## 2.2   Data

The database of the analysis is consistent with the data described in Knevels et al. (2020) and Maraun et al. (2022). While detailed information on the landslide inventory, the sampling design and the landslide susceptibility model can be found in Knevels et al. (2020), model extensions and applications that account for soil moisture and environmental changes, respectively, were presented in Maraun et al. (2022). Similar to the above-mentioned studies, we used a target resolution of 10 m × 10 m for our analysis. Here we will present the database only briefly giving the details relevant for our model construction, and we refer

the reader to the cited publications for further detail.

### 2.2.1   Base Data

For the analysis, different climate, land surface and landslide data were provided by various sources. As climate data, precipitation and soil moisture data was used. INCA (Integrated Nowcasting through Comprehensive Analysis, 1 km × 1 km) precipitation was provided by the Austrian Central Institute for Meteorology and Geodynamics. Furthermore, precipitation

was aggregated to obtain accumulated five-day rainfall (in mm) and maximum three-hour rainfall intensity (in mm h$^{-1}$) on the landslide failure day, respectively. The soil moisture data was derived using HRLDAS v4.1 (high-resolution land data assimilation system, Chen et al. 2007) with a 1 km × 1 km spatial and an hourly temporal resolution for the 2004–2014 period, and was provided by Maraun et al. (2022) (technical details in Schaffer 2021). As land surface data, an airborne LiDAR-derived high-resolution digital terrain model (HRDTM, 1 m × 1 m), a geological basemap, and LULC data (distinguishing forest types:

no/broadleaf/mixed and conifer forest) were provided by the GIS department of Styria and JOANNEUM RESEARCH, respectively. Based on the HRDTM suitable land surface variables were derived (Sect. 3.1).

As landslide data, we used landslides that occurred during the extreme rainfall events, which were initially mapped by the Institute of Military Geoinformation and the Geological Survey of Austria in 2009, and in 2014 by the Department of Hydrology, Resources and Sustainability of the Styrian Government, respectively. Knevels et al. (2020) analysed and filtered

these datasets, and compiled a quality-controlled landslide inventory comprising 626 landslides (487 for 2009 and 139 for 2014) in total, which we classified as earth and debris slides with possible transitions to complex slide flows after Cruden and Varnes (1996) (refer to Table A2 in Knevels et al. (2020) for more information).





### 2.2.2 Environmental Change Simulations

In order to project landslide susceptibility patterns based on past and future environmental conditions, we obtained event
storylines in pre-industrial and future climates (Maraun et al., 2022) as well as a future LULC scenario compiled from various
sources. In an event storyline approach, the emphasis is placed on a qualitative understanding and plausibility of driving
factors involved in an event, and thus the physically self-consistent unfolding of past, or plausible future events or pathways is
examined (Shepherd et al., 2018).

For climate change, the focus was on simulations of the 2009 rainfall event in the present climate, as well as of its character-
istics in pre-industrial and future climates. The pre-industrial ("past") climate is understood here as a counterfactual (present)
climate, i.e. without climate change (from here on referred to as NO-CC for ´climate with no climate change´). The simulations
were based on the regional climate model (RCM) Consortium for Small-scale Modelling (Rockel et al., 2008) with a spatial
resolution of $3\,km \times 3\,km$ and covered the eastern Alpine region (Fig. S1 in Supplementary Material). The RCM boundary
conditions were obtained from the integrated forecast system of the European Centre for Medium-Range Weather Forecasts
(Bechtold et al., 2008). A spin-up was run (1 October 2008 to 20 June 2009) to ensure a balanced soil-moisture field. Based
on the spin-up, a ten-member ensemble simulation of the actual event ending at 28 June 00:00 UTC was computed ("present-
day"). Storylines for NO-CC and future conditions were simulated based on boundary conditions modified by changes from
four global climate models (GCM) of the Coupled Model Intercomparison Project (Taylor et al., 2009) accounting for the
representative concentration pathways with a radiative forcing of $8.5\,W\,m^{-2}$ (RCP8.5, i.e. high emission scenarios; with ten
ensemble members each): IPSL-CM5A-MR (IPSL), HadGEM2-CC (HadGEM), GFDL-ESM2m (GFDL), and MIROC-ESM
(MIROC). For the future simulations, the RCM boundaries were modified by imposing changes derived from GCMs, represent-
ing the difference between typical conditions during weather events comparable to the 2009 event in warmer future climates
and present climate. The changes were derived from events occurring in the periods 2071–2100 and 1975–2004 under the
RCP8.5 scenario and rescaled to selected global warming levels (Maraun et al., 2022). Specifically, we considered 0.5 K (Paris
agreement; PARIS), 3 K (business-as-usual) and 4 K warming (worst case). For the NO-CC simulations, boundary conditions
were obtained by scaling the future climate change signal down to $-1\,K$ cooling. Please refer to Maraun et al. (2022) for more
details on the climate simulations.

Since all conducted simulations showed a positional bias, a delta change approach was applied for each hydrometeorological
variable (i.e. precipitation and soil moisture). Following Maraun et al. (2022) the delta change factors were estimated for the
simulated hydrometeorological values as ratios of differences in areal averages within domains where the climate change signal
was assumed to be constant (i.e., NO-CC to present, present to future, $10 \times 10$ ensemble-members, i.e. 100 pairs per storyline).
In contrast to Maraun et al. (2022), in this study, for soil moisture, the region was expanded to fit the actual target domain (i.e.,
southeast Styria vs. Feldbach region, Fig. S1 in Supplementary Material).

A LULC scenario for the future was developed jointly with the Forestry Directorate and District Forestry Authority as
regional and local stakeholders. The present-day Syrian forest has a characteristic structure of small-scale changes of different
forest types, but with a high percentage of spruce (around 58 %, BFW 2019). Rising temperatures and summer dryness may





lead to a higher vulnerability to disturbances such as pathogens and forest pests, including bark beetles (Kolström et al., 2011; Jandl, 2020). Adopting active forest management in the developed future LULC scenario, coniferous forest was replaced by climate resilient mixed forest. Additionally, present-day agricultural land in unfavourable topography (e.g., slopes steeper than
20 °) were simulated as being transformed to mixed forest in the future. For the NO-CC, the present-day LULC was used as we were only interested in the climate signal.

### 2.2.3  Data for Landslide Susceptibility Predictions

For the spatial prediction of landslide susceptibility, the creation of a prediction dataset with one layer for each variable (i.e., stack of raster layers) was required. Since precipitation and soil moisture varied during the landslide-triggering period (22–
25 June 2009) and landslide failure dates for the simulated NO-CC and future are unknown, we extracted process-related aggregated features for the considered period. Specifically, for each grid cell we determined the maximum three-hour rainfall intensity, and we took the maximum five-day rainfall. For soil moisture, we used the maximum value on the day prior to the beginning of the corresponding five-day rainfall aggregation. We downscaled the climate data to our target resolution using the inverse distance weighting method (*power = 3, maximum number of neighbours = 16*).

## 3  Methods

### 3.1  Landslide Susceptibility Model

Our analysis built on the landslide susceptibility model of Maraun et al. (2022), which was a generalised additive model (GAM). A GAM is a semi-parametric extension of a generalised linear model (GLM) since it has the ability to model non-linear relationships by automatically fitting transformations, or so-called component smooth functions (Hastie and Tibshirani,
1986; Wood, 2017). The additive structure of a GAM allows common model diagnostics (e.g., predictor-response relationship, variable importance, odds ratios; Brenning 2012), and thus GAMs have become popular in landslide susceptibility studies in recent years (Petschko et al., 2014; Knevels et al., 2020).

For the landslide susceptibility analysis, we linked predisposing and triggering factors to landslide occurrences. Predictor variables were land surface variables (convergence index of 100 m and 500 m, plan and profile curvature, logarithmic D-Infinity
flow accumulation, normalised height, slope angle, slope angle of catchment area, North- and West-exposedness, topographic position index, and topographic wetness index), hydrometeorological variables (soil moisture, five-day rainfall and maximum three-hour rainfall intensity), geology, and forest type (representing LULC).

We developed landslide susceptibility models with different model settings, some of which have been published before (Table A1 in Appendix A). As a new model, we decided to explicitly account for residual spatial autocorrelation using a low rank
Gaussian process smoother (GP, Wood 2017) as an additional predictor (GAM-Spatial). This feature is often missing in landslide susceptibility modelling, which may result in residual patterns that are unaccounted for by the model and may introduce errors in predictions (Reichenbach et al., 2018). Furthermore, we kept the influence of the five-day rainfall variable constant




beyond 80 mm ("top-coded") to counteract a physically implausible predictor-response relationship beyond this threshold (GAM-SM+TC in Maraun et al. 2022). In other model settings, the five-day rainfall variable was not modified (i.e., GAM-SM,

Maraun et al. 2022; cf. Fig. A1 Appendix A), and soil moisture was not included as model predictor (i.e., GAM-Co, Knevels et al. 2020).

We used a GAM-Spatial implementation that incorporates residual spatial autocorrelation through the GP representation (Fahrmeir et al., 2013; Wood, 2017). The hyperparameters necessary for the GP implementation are a range ($\Phi$) and correlation function, and were estimated based on the residuals of GAM-SM+TC. We followed Simpson (2018) in iteratively searching

for a suitable range (from 10 to 1000 m) and correlation function (spherical, exponential, Gaussian, and Matérn, the latter with $\kappa$ values of 1.5, 2.5 and 3.5). The models' restricted maximum likelihood (REML) score was assessed to identify optimal hyperparameters (i.e., lowest REML score), which were then applied to fit the GAM-Spatial model with a GP. Additionally, the effective range, at which the residual autocorrelation drops below 0.05, was calculated using *StempCens* in R (*EffectiveRange* with $cor = 0.05$, Valeriano et al. 2020).

For model assessment, we used well-established diagnostic tools. The model performance was assessed using a five-fold spatial cross-validation (SpCV) with five repetitions, and measured using the area under the receiver operating characteristic curve (AUROC) (Brenning, 2012; Knevels et al., 2020). We ensured that the training and validation data were identical to those in Knevels et al. (2020) to achieve a fair comparison of the landslide susceptibility models. The AUROC values were interpreted following Hosmer et al.'s (2013) interpretation guide. To assess variable importances, we calculated the model's mean

decrease in deviance explained (mDD, %) after removing the respective variable from the model (Knevels et al., 2020). Larger mDD values indicate greater explanatory power. Predictor-response relationships were analysed visually using transformation functions and quantitatively using odds ratios (OR). An OR represents the chance that an outcome happened given a specific exposure, compared to odds of the outcome under a reference exposure (Szumilas, 2010). An OR greater than one means an exposure with higher odds of landslide occurrence while an OR lower than one is associated with lower odds of landslide

occurrence, while accounting for the other variables in the model (OR = 1 means no association to the exposure). In contrast to Knevels et al. (2020), the variable importance and predictor-response relationships were assessed based on GAMs using the complete data set (GAM-Spatial, GAM-SM+TC, GAM-SM), i.e. not on subsets of the SpCV models (GAM-Co).

We focused on highly susceptible areas for analysing the uncertainty cascade in storylines of the landslide susceptibility. The area of high landslide susceptibility was defined using the thresholding approach of Petschko et al. (2014), according to

which 70 % of the observed landslides fall into that class; low and medium susceptibility account for 5 % and 25 % of observed landslides.

We used the free and open-source computing environment R (R version 4.1.0, Team 2021) with the GAM implementation in the *mgcv* package (Wood, 2017).

## 3.2 Modelling Landslide Susceptibility in a Changing Environment

Landslide susceptibility is commonly considered stationary (or invariant in time; Fell et al. 2008; Gariano and Guzzetti 2016; Reichenbach et al. 2018). However, this assumption is often violated in space (e.g., anthropogenic land use changes, Reichen-



bach et al. 2014) and time (e.g. "follow-up" landslides, Samia et al. 2017). In this study, we assessed NO-CC and future landslide susceptibility by considering the LULC and hydrometeorological variables as time-varying predictors, and therefore modified their values according to the storylines. While the hydrometeorological predictors were multiplied with the

corresponding delta change factor, the LULC data was replaced with the developed scenario only for the future. The other predictors (land surface variables and geology) were considered as invariant in time.

To estimate changes in landslide susceptibility between a storyline and the present day, we applied two approaches: First, we calculated the relative change in the areal extent of a susceptibility class (in % and percentage points, pp) to quantitatively derive the potential change in the area at risk. And second, we expressed the effect size of change using the average change in

the odds of landslide occurrence of a susceptibility class as $OR_{class}$ for those locations which fall into identical susceptibility classes in both predictions:

$$OR_{class} = \exp\left(\frac{1}{n}\sum_{i=1}^{n}(logit_{story}(i) - logit_{ref}(i))\right) \tag{1}$$

where $logit_{story}(i)$ and $logit_{ref}(i)$ denote the logits of landslide occurrence probabilities at a specific location $i$ and susceptibility class (low, medium, or high) of the storyline and reference scenario ("present day"), respectively, and $n$ as all locations

of the same landslide susceptibility class in both scenarios.

### 3.3 Uncertainty Cascade in Landslide Predictions

In integrated modelling, there are various sources of uncertainty affecting the predicted outcomes. We quantified the uncertainty cascade in storylines of landslide susceptibility by accounting for climate model uncertainty and landslide model parametric uncertainty (or sampling variability).

For the climate model uncertainty we assessed within-event internal model variability and scenario uncertainty for each hydrometeorological variable. The applied single event storyline approach includes a particular realisation of internal climate variability, thus the established storylines are conditioning on it (Doblas-Reyes et al., 2021). By using estimates from a climate model ensemble with different realisations of internal variability, the uncertainty related to conditional internal climate model variability can essentially be removed (Maraun et al., 2022). In this study, we explicitly analyse within-event internal climate

model variability, and therefore address the question how the event could locally unfold in the present climate as well as cooler and warmer climates. Regarding the climate signal or forced change, the influence of within-event internal climate model variability can be accounted for by averaging across the ensemble members of a climate models' storyline (Maraun et al., 2022).

For the within-event internal climate model variability, we estimated the 2.5th and 97.5th percentiles from the delta change

factor distribution (i.e. 100 pairs per storyline) describing the lower and upper bound of local climate model variability; the mean delta change factor represents a models' climate signal (or forced change). The estimated delta change factors were subsequently offset against the corresponding values of the hydrometeorological predictor. For the climate scenario uncertainty,


the spread of the predicted outcomes using the climate signals of all climate models within a scenario (NO-CC, PARIS, 3 K and 4 K warming) was calculated.

The landslide models' parametric uncertainty is associated with measurement errors, sampling errors and variability, misclassification of data and surrogate data weaknesses (EPA, 2019), and, from a frequentist perspective, commonly expressed using confidence intervals. A common approach to quantify parametric uncertainty is bootstrapping (e.g., Brenning et al., 2015). However, in the framework of the *mgcv::gam*, this approach has limitations when penalties are present and some observations are sampled twice. This may cause undersmoothing and is therefore not recommended (Wood, 2017). However,

the manner the *mgcv* GAM is implemented can be viewed as 'empirical Bayes' (Wood, 2017). This allows the calculation of *pointwise* Bayesian credible intervals for the estimated model parameters using the standard error derived from the Bayesian posterior covariance matrix (Wood, 2017; Simpson, 2018). The pointwise Bayesian credible intervals have good frequentist coverage probabilities when averaged across the function domain, but with over- and under-coverage in some parts (Nychka, 1988; Marra and Wood, 2012). In contrast, a *simultaneous* Bayesian credible interval that contains the entire true function for

a given credible level $(1-\alpha)$, can be achieved by posterior simulation from the posterior distribution of the model coefficients (Simpson, 2018); this is the approach we adopt in this study.

     To simulate GAM coefficients, a Gaussian approximation to the posterior for the coefficients is a computationally very efficient approach, but not recommended for spatial models with a logit link in the *mgcv* framework (Wood, 2015). Therefore, we used a simple Metropolis Hastings (MH) sampler with random walk proposals as implemented in the *mgcv::gam.mh*

function, but also calculated a Gaussian approximation for comparison purposes (*mgcv::rmvn* function). The MH function reports two types of acceptance rates: a fixed acceptance rate and a random walk acceptance rate (refer to Wood 2015 for more details). The simulations of the MH sampler are controlled by hyperparameters that need to be tuned to achieve an optimal proposal distribution. A proposal distribution is optimal for a random walk acceptance rate of 23.4 %, while high values are to be achieved for the fixed acceptance rate (Roberts et al., 1997; Wood, 2015). Thus, the optimum of the hyperparameter tuning

was selected by minimising the sum of the absolute difference of both acceptance rates to their optimal rates. For the tuning, we set the initial burn-in period to 5000 simulations, the number of (actual) "simulated GAMs" to 10,000, and optimised *rw.scale* (random walk scale factor for posterior covariance matrix) and *t.df* (degrees of freedom of the t distribution) using a grid search. We defined a regular grid ranging from 0.005 to 0.02 for *rw.scale* (step size of 0.0005, i.e. 31 total steps) and from 25 to 10,000 (step size of 25, i.e. 400 total steps) for *t.df*, respectively (in total 12,400 executions).

Furthermore, the simulation from the posterior distribution allows to derive the so-called critical value for each nonparametric transformation function (Ruppert et al., 2003; Simpson, 2018). The critical value is a factor which modifies the pointwise Bayesian credible interval to achieve the simultaneous Bayesian credible interval for a given credible level. A critical value greater than the coverage factor for a given credible level means an underestimation of the true function for the pointwise interval compared to the simultaneous interval (e.g. critical value of 2.5 vs. coverage factor of 1.96 for 95 % credible

level,  28 % underestimation; Simpson 2018), and large critical values may indicate particularly uncertain predictors.

     For the uncertainty cascade, landslide susceptibility prediction uncertainty intervals were estimated as described in the following. First, we classified the predictions of each of the simulated GAM-Spatial models into (low, medium, and) high





landslide susceptibilities. And second, we derived the 2.5th and 97.5th percentiles as lower and upper uncertainty intervals of the corresponding storyline from the resulting distribution of the high landslide susceptibility area (i.e., 3 x 10,000 values

of high landslide susceptibility for mean and percentiles of delta change factor distribution). Finally, to compare the different sources of uncertainty in the uncertainty cascade, we calculated the ratio of the uncertainty spread of within-event internal climate model variability and scenario uncertainty (i.e., spread of climate signals), respectively, to the parametric uncertainty spread of the landslide model. The joint uncertainty distribution of the landslide models' parametric uncertainty and within-event internal climate model variability is here referred to as the storyline uncertainty.

## 290 4 Results

### 4.1 Climate Change Uncertainty

We identified different patterns of delta change factor distributions depending on the considered hydrometeorological variable and the storyline (Fig. 2).

The mean delta change factors (i.e., climate signals) for five-day rainfall and maximum three-hour rainfall intensity were

generally lower in the NO-CC and higher in the future relative to present-day climate (Table A3 in Appendix A). In the future climate, with the exception of the GFDL model, the mean increase ranged from 3 % (IPSL, PARIS) to 34 % (MIROC, 4 K warming) for five-day rainfall and from 4 % (HadGEM, PARIS) to 61 % (MIROC, 4 K warming) for maximum three-hour rainfall intensity. In the NO-CC, with the exception of the GFDL model, the changes were negative on average (i.e., factors < 1). Specifically, mean decreases were between 3 % (IPSL) and 6 % (HadGEM) for five-day rainfall, and ranged from 10 %

(HadGEM) to 13 % (MIROC) for maximum three-hour rainfall intensity. The averaged delta change factor of GFDL indicated projected changes of ±1 % in both meteorological variables in NO-CC as well as future climates. For soil moisture, the models projected on average drier soils in future and wetter soil in NO-CC relative to present-day climate, with the exception of HadGEM. IPSL in 3 K warming projected the driest soils (−16 %, on average), and it also projected the wettest soil conditions overall (+2 %) in NO-CC, on average. Following Maraun et al. (2022), the hydrometeorological storylines for the future can

thus be summarised as as (much) heavier rain (HadGEM, MIROC, IPSL) and (much) drier soil (IPSL, GFDL, MIROC), although there were also some neutral projections (rain: GFDL; soil: HadGEM; Table A2 in Appendix A).

Regarding the 2.5th and 97.5th percentiles of the storylines' delta change factor distributions (i.e., within-event internal climate model variability), we discovered opposed delta change factors in the GFDL storylines (Table A3 in Appendix A). Similarly, in the NO-CC and PARIS scenarios for five-day rainfall, all other climate models had contrasting delta change

factors, while for the maximum three-hour rainfall intensity this was only the case for HadGEM and IPSL in PARIS. Regarding the percentiles of the soil moisture storylines, only coherent delta change factors were found, yet some climate signals were weak (e.g., nearly 1 for HadGEM).


**Figure 2.** Delta change factor distribution for each storyline; variability corresponds to individual model simulations. Note: Scales of y-axes differ between scenarios. Boxplot modifications: The middle hinge shows the mean. The lower and upper whiskers represent the 2.5th and 97.5th percentiles, respectively. Potential outliers are not plotted.

## 4.2 Landslide Susceptibility Model

We established a spatial GAM using an optimised GP smoother. The spatial structure of GAM-SM+TC residuals was best described by a Gaussian correlation function with an effective range of 524 m (Φ of 303 m). Nevertheless, the other correlation models achieved nearly identical REML scores with varying effective ranges (582 m to 710 m, Table A4 in Appendix A and Fig. S2 in Supplementary Material). The following subsections describe the model assessment (Sect. 4.2.1) and the posterior simulation of the coefficients (Sect. 4.2.2) for GAM-Spatial. Detailed results for the other model settings (GAM-Co and GAM-SM+TC) are included in the Supplementary Material (Fig. S3).

**Figure 3.** Model diagnostics for GAM-Spatial. (a) Top five most important variables sorted by mean decrease in deviance explained [%]. For an overview of all input variables, refer to Table A6. (b) Comparison of model performances (folds-based). (c) Comparison of predictor-response relationships of LULC variables using odds ratios. (d) Comparison of predictor-response relationships of hydrometeorological variables. Note: The y axes in (d) are plot-dependent. Estimates and predictor-response relationships for GAM-Co are based on SpCV models in Knevels et al. (2020). In grey: 95 % pointwise Bayesian credible intervals.

### 4.2.1 Assessment of the Landslide Susceptibility Model

GAM-Spatial achieved an outstanding discrimination capability with mAUROC of 0.94 (Fig. 3a, Table A5 in Appendix A). The model's five most important variables in decreasing order were five-day rainfall (10.8 % mDD), forest type (8.8 % mDD), the spatial GP smoother (6 % mDD, longitude and latitude), slope angle (5.5 % mDD) and profile curvature (1.1 % mDD) (Fig. 3b). The hydrometeorological variables maximum three-hour rainfall intensity (0.9 % mDD) and soil moisture ranked sixth and ninth (0.6 % mDD), respectively (Table A6 in Appendix A).


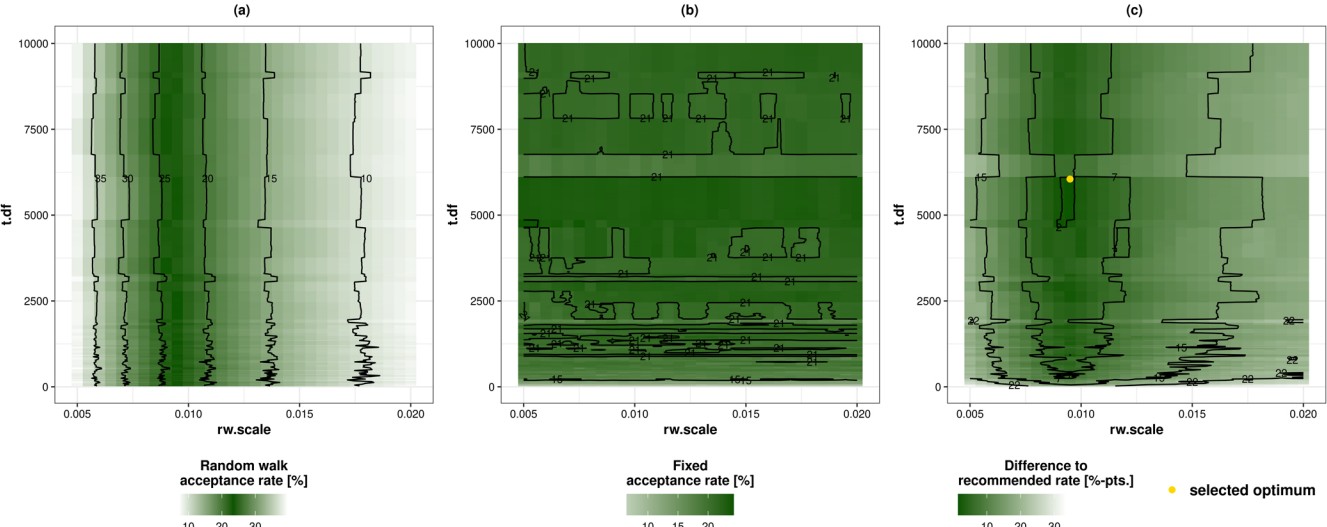

**Figure 4.** Optimization results for acceptance rates of *mgcv::gam.mh* using different hyperparameter values for *rw.scale* (random walk scale factor for posterior covariance matrix) and *t.df* (degree of freedom for t proposal). (a) Random walk acceptance rate. (b) Fixed acceptance rate. (c) Sum of absolute differences between acceptance rates to optimal rate. The optimal rate for random walk acceptance is 23.4 % according to Roberts et al. (1997), and the highest possible rate for fixed acceptance, respectively.

Regarding the predictor-response relationships, for the LULC variable, the chances (i.e., odds) of landslide occurrence were up to 0.04 times as low in forested areas as in non-forest areas (Fig. 3c). For five-day rainfall, the modelled chances rise as rainfall increases up to 80 mm, where a plateau was reached due to the top-coding of this variable. For maximum three-hour rainfall intensity and soil moisture, we identified a linear relationship with higher odds of landslide occurrence for higher

rainfall and soil moisture values, respectively, while accounting for the other variables in the model (Fig. 3d).

Overall, the inclusion of the spatial GP smoother had little impact on the model discrimination capability, relative importance of variables or on the modelled relationships compared to other model settings (Fig. S3 in Supplementary Material).

### 4.2.2 Hyperparameter Tuning for Posterior Simulation

The hyperparameter tuning for the posterior simulation allowed us to identify an optimal region at *rw.scale* = 0.0095 and *t.df*

between 6050 and 6100 (Fig. 4c). For the random walk acceptance rate, we discovered a non-linear relationship of *rw.scale* to the random walk acceptance rate describing higher acceptance rates at lower *rw.scale* values, while changing *t.df* values had no effect (Fig. 4a, Fig. A2 in Appendix A). The fixed acceptance rate, in contrast, increased with higher *t.df* values and reached a plateau at *t.df* values around 2500 (Fig. 4b, Fig. A2 in Appendix A). Considering the optimal rates, we identified three optima with identical *rw.scale* of 0.0095, random walk acceptance rate of 23.24 % and fixed acceptance rate of 22.95 %, and nearly

identical *t.df* values (6050, 6075, 6100; overall difference of 1.47 pp, Fig. 4c). Therefore, we simulated coefficients based on the optimal hyperparameter values corresponding to the lowest *t.df* value (i.e., 6050).





The estimation of the critical values for the non-parametric transformation functions according to Simpson (2018) revealed a general agreement of the Gaussian approximation (*mgcv::rmvn*) and the simple Metropolis Hastings sampler (*mgcv::gam.mh*) (Table A7). For GAM-Spatial's five most important variables, the critical values ranged between 3.7 (longitude, latitude) and 345  37.85 (five-day rainfall), which corresponds to intervals that are approximately $\pm 88\,\%$ to $\pm 1{,}831\,\%$ wider than the equivalent across-the-function intervals for the 95 % credible level.

### 4.3 Uncertainty Cascade in Landslide Predictions

The landslide susceptibility predictions of the simulated GAM-Spatial models were classified into low, medium and high susceptibility (Petschko et al., 2014). The thresholds to discriminate low from medium and medium from high landslide sus- 350  ceptibility were 0.16 and 0.57 on the probability scale, respectively. In the following, we focus on the storylines of highly susceptible areas and their uncertainties (Fig. 5).

Regarding the present-day landslide susceptibility, around 4.9 % of the study area was highly susceptible to landslides (Fig. 5a, Table A8 in Appendix A). In the NO-CC scenario, high susceptibility was generally less common (4.4–4.8 %), with the exception of GFDL, which showed a slightly higher share (5.1 %).

The projected future signal, in contrast, was less coherent. In PARIS, the mean signal was similar to present-day susceptibility ($-0.2$ pp [percentage points difference in area] for GFDL and $+0.2$ pp for HadGEM) while the mean changes in 3 K and 4 K warming were substantial. In 3 K and 4 K warming, GFDL and IPSL projected a decrease in highly susceptible area by up to $-1.3$ and $-2.2$ pp ($-27$ and $-45\,\%$), respectively, while HadGEM and MIROC showed the opposite signal with increases of up to $+1.7$ and $+0.8$ pp ($+35$ and $+16\,\%$), respectively. The mean signals of climate models in a 3 K warmer world were 360  comparatively lower than in 4 K. Additionally, the considered LULC scenario resulted in a generally smaller extent of highly susceptible areas (up to 0.5 pp).

Regarding the OR of the storylines relative to present-day landslide susceptibility within the high-susceptibility class (see eq. 1), the pattern is similar to the change in susceptible area, but more coherent (Fig. 5b, Table A9 in Appendix A). The chances of landslide occurrence in the NO-CC scenario were 0.91 (HadGEM) to 0.98 times (IPSL) as high as in the present-day settings 365  (with the exception of GFDL with OR of 1.05), based on the modelled climate signals. For the future, while in PARIS the ORs showed small effect sizes (ORs between 0.97 for GFDL to 1.04 for HadGEM), in 3 K and 4 K warming the effect sizes were medium to large. In the worst-case storyline, the chances of landslide occurrence in highly susceptible areas were 1.37 times as high as at the present day (i.e., for HadGEM in 4 K warming), while it was 0.60 times as high in the best-case storyline (i.e., for IPSL in 3 K warming).

Regarding the storyline uncertainty (Fig. 5a, Table A7 in Appendix A), for present-day landslide susceptibility, the uncertainty spread from $+1.4$ pp ($+29\,\%$ area change) to $-1.1$ pp ($-22\,\%$), respectively. While present-day predictions were not affected by within-event internal climate model variability and thus reflected only landslide model uncertainty, the other storyline uncertainties accounted for both. The largest uncertainty interval for the NO-CC covered 3.0 pp (GFDL). For the future, we identified larger uncertainty intervals with increased warming levels (largest intervals: PARIS: 3.2 pp for HadGEM, 3 K 375  warming: 4.4 pp for MIROC, 4 K warming: 5.4 pp for MIROC).

**Figure 5.** Uncertainty assessment results for high landslide susceptibility. (a) Area of high landslide susceptibility with 95 % CIs are based on within-event internal climate model variability and landslide model uncertainty. (b) OR of landslide occurrence in highly susceptible areas relative to present-day landslide susceptibility. (c) Ratio of within-event internal climate model variability and climate scenario uncertainty, respectively, to landslide model uncertainty.



Regarding the contributions of the uncertainty cascade in the storylines, the uncertainty introduced by within-event internal climate model variability is around 0.13 (IPSL in 3 K warming) to 0.35 (HadGEM LULC in 4 K warming) times as large as parametric landslide model uncertainty (Fig. 5c, Table A10 in Appendix A). Instead, the ratio of climate scenario uncertainty to parametric landslide model uncertainty showed a distinct pattern depending on the scenario: For NO-CC and PARIS, the

uncertainty introduced by the climate models was lower relative to parametric landslide model uncertainties (ratios between 0.28 and 0.34). Instead, for 3 K warming, the climate scenario uncertainty was generally larger than the parametric landslide model uncertainty with ratios between 1.05 (IPSL) and 1.46 (GFDL LULC) (excluding MIROC with a ratio of 0.97). For 4 K warming, while the climate scenario uncertainty was larger than the landslide model uncertainties for GFDL and GFDL LULC (ratios of 1.23 and 1.38, respectively), for the other models the ratios were equal to (i.e., HadGEM LULC) or lower than 1

(e.g., ratio of 0.77 for MIROC).

## 5 Discussion

### 5.1 Landslide Susceptibility in a Changing Environment

We identified different trends of projected landslide susceptibility for future storylines relative to present-day, while for the NO-CC storylines (i.e. pre-industrial, counterfactual climate), landslide susceptibility was generally estimated to be lower (exception: GFDL). For projected future changes in hydrometeorological conditions (i.e., climate signals), the storyline of

*much heavier rain* showed the strongest increase in affected highly susceptible area (+35 %) with additionally higher chances of landslide occurrence relative to present-day susceptibility (OR of 1.37, HadGEM in 4 K). However, the effect of *much drier soil* may compensate for *heavier rain*, and thus reduce the affected area (−45 %) and the chance of landslide occurrence (OR of 0.6, IPSL in 3 K). The projected changes in highly susceptible areas and their magnitudes were in general agreement with the findings of Maraun et al. (2022), who used a small part of our study area (Feldbach region, 95th percentile for high susceptibility

thresholding). For the worst- and best-case storylines, Maraun et al. (2022) found changes of +45 % (4 K warming) and −37 % (3 K warming) in highly susceptible area, and ORs of 1.66 (4 K warming) and 0.8 (3 K warming) in landslide susceptibility in those areas.

For other regions, some authors projected an increased landslide occurrence or affected area that is attributable to climate
change (Jaedicke et al., 2008; Lee, 2017), while other authors found more complex patterns that depend on the seasonal period and location under consideration (Ciabatta et al., 2016; Gariano et al., 2017; Rianna et al., 2017). Projected increases in slope failure probability amounted to approximately 25 % for mountainous areas in Norway (for 2050; Jaedicke et al. 2008) or 40 % for the Umbria region in central Italy (for 1990–2013 to 2070–2099 under RCP8.5; Ciabatta et al. 2016), and an increase in susceptible area by around 32 % was projected for southwestern Taiwan (for 2090; Lee 2017). With a focus on

seasonal differences, Ciabatta et al. (2016) projected increases in landslide occurrence mainly in winter, while in the warm and wet season, very low soil moisture (Ciabatta et al., 2016) and increased evaporation (Rianna et al. 2017; for 2071-2100 under RCP8.5) might even improve slope stability. Analysing spatial patterns in Calabria, southern Italy, Gariano et al. (2017) identified for around 27 % of the municipalities a significant increase in rainfall-events with landslides (western part of the





region and along the main mountain chains), while for 38 % of the municipalities a reduction was estimated (for 1981–2010
to 2036–2065 under RCP8.5). Therefore, we agree with Schlögel et al. (2020) that the projected interplay between a natural
hazard and climatic changes is still a challenging task, especially if multiple triggers and locally driven ground responses are
present.

Regarding changes in LULC, our findings suggest that active LULC management and afforestation may have a beneficial
effect on landslide occurrences (−0.5 pp in highly susceptible areas). Such effects were also reported by other authors (Picarelli
et al., 2017; Pisano et al., 2017; Gariano et al., 2018).

## 5.2 Uncertainties of a Changing Environment

The analysis of uncertainties in landslide risk management under environmental change is an important, yet often neglected task
(Reichenbach et al., 2018). In particular, the analysis of uncertainty cascades in integrated modelling is still an open research
gap with a great potential to understand and subsequently reduce uncertainty sources (Refsgaard et al., 2016; Kirchner et al.,
420 2021).

In our analysis, we could successfully quantify uncertainty in landslide susceptibility predictions by accounting for para-
metric uncertainty in the landslide model and climate change uncertainty (within-event internal model variability and scenario
uncertainty). However, we found the estimated uncertainties in predictions to be generally high, which is not unusual for an
uncertainty cascade (Refsgaard et al., 2016). Furthermore, we discovered the tendency towards higher uncertainty with in-
creased warming level for both parametric uncertainty and within-event internal climate model variability. Uncertainty from
within-event internal climate model variability was much lower than parametric uncertainty (ratio of around 0.25). Addition-
ally, parametric uncertainty even exceeded scenario uncertainty for specific climate models (e.g., ratio of 0.77 for MIROC in
4 K warming). Other authors reported similar uncertainty components in the context of hydrological modelling under climate
change (Bastola et al., 2011; Refsgaard et al., 2016).
430 In the following subsection the uncertainties in climate change modelling (Sect. 5.2.1) and landslide susceptibility modelling
(Sect. 5.2.2) are further discussed.

### 5.2.1 Climate Change Uncertainty

In climate change modelling, there were different sources of uncertainty. We addressed uncertainties in climate sensitivity or
global climate response uncertainty by conditioning the climate results on global warming levels, and thus we were capable
435 of approximately removing this uncertainty (Chen et al., 2021; Maraun et al., 2022). Local climate response uncertainties
were represented by the simulation of storylines. Furthermore, climate change projections are generally influenced by scenario
uncertainty and internal variability (Stainforth et al., 2007). Scenario uncertainty was accounted for by considering landslide
susceptibility conditional on different levels of global warming. Internal variability mainly arises from large-scale circulation
(Shepherd, 2014), which was kept fixed and thus removed. Uncertainties related to the influence of within-event internal
440 variability on local changes can effectively be removed (i.e. by averaging across the ensemble members; Maraun et al. 2022),





but were explicitly addressed in this study by accounting for the delta change factor distribution estimated on all possible pairs of a climate model's simulation.

We discovered that in our case study, the climate signal of the hydrometeorological variables showed generally more and intensified precipitation and a drier soil in future storylines (in contrast to the NO-CC storylines), which was also reported for extreme rainfall events in other regions (Ciabatta et al., 2016; Rianna et al., 2017; Olefs et al., 2021). However, regarding the within-event internal climate model variability, contrasting delta change factors were identified mainly for NO-CC and PARIS storylines, and especially for the variable five-day rainfall. While for the NO-CC the contrasting delta change factors showed that there is a low but non-zero probability that a similar event without climate change could be locally stronger than the actual event in present climate, it means for the targeted 0.5 K warming limitation (Paris agreement) that an event may unfold locally weaker than under present climate. Furthermore, the contrasting delta change factors of the GFDL climate model (in all storylines and for both meteorological variables) clearly indicates the importance of a careful selection of multiple, reliable climate models. The availability of multiple storylines ultimately avoids an under- or overestimation of the projected landslide activity (Gariano et al., 2017).

### 5.2.2 Landslide Susceptibility Model Uncertainty

In a climate change context, uncertainties from a landslide model arise from model structure and parametric uncertainty under extrapolation (Maraun and Widmann, 2018). While parametric uncertainty results from the finite sample and the fact that the observed landslide distribution was a realisation of a stochastic process, structural uncertainty comprises the presence of physically plausible climatic predictor-response relationships for environmental change conditions (Maraun et al., 2022). Additionally, by calibrating the landslide model on two rainfall events (2009 and 2014, with different hydrometeorological conditions), we adjusted the model as far as possible for extrapolation purposes.

For the assessment of the parametric uncertainty, we simulated from the posterior distribution of coefficients using a Metropolis Hastings sampler. For the most important predictor-response relationships, we identified simultaneous Bayesian credible intervals that were by far larger than the corresponding pointwise intervals at a 95 % credible level ($\pm 88 \%$ to $\pm 1{,}831 \%$), which explained the generally high parametric uncertainty. Especially, the predictor-response relationship of five-day rainfall appeared very uncertain (critical value of 37.8, Table A7 in Appendix A), while the simultaneous Bayesian credible intervals of the other hydrometeorological variables were only marginally wider than their pointwise counterparts. Furthermore, the top-coding of the variable at 80 mm additionally increased the parametric uncertainty (cf. critical value of 22.7 for GAM-SM, i.e. no top-coding of five-day rainfall). Moreover, with the application of higher and lower delta change factors, respectively, in the higher warming levels, predictor values shifted towards the bounds of the training distribution, where data tends to be sparse and, therefore, uncertainty increases.

For the reduction of structural uncertainty, we applied a physically plausible statistical landslide model that allowed us to assess the landslide response in a changing climate by varying hydrometeorological predictors. Especially, the use of moisture-related predisposing factors is an important, yet an often missed requirement in event-based landslide modelling (Bogaard and Greco, 2018). For the variable five-day rainfall, we top-coded the value range at 80 mm to remove implausible fluctuations





beyond that threshold (Maraun et al., 2022). A similar relationship of unchanged slope conditions with higher rainfall was
also identified in Lee (2017) (described as Weibull cumulative distribution function). For the soil moisture variable, we identi-
fied a linear predictor-response relationship showing higher landslide occurrence probability with higher soil moisture values.
However, other authors reported a more complex soil moisture relationship due to the interplay with temperature (e.g., desic-
cation cracking; Tang et al. 2018; Debele et al. 2019) or soil texture (coarse-grained vs. fine-grained soil, Rianna et al. 2016),
which affects geomechanical characteristics and thus slope stability. Besides, the soil moisture variable had only a moderate
explanatory power (0.6 % mDD), although antecedent soil moisture conditions had likely contributed to the landslide event's
severity (Hornich and Adelwöhrer, 2010). We suggest that the thematic and spatial resolution of the soil moisture estimated
by HRLDAS (Schaffer, 2021; Maraun et al., 2022) was too coarse to capture the "true" soil moisture in the field, and that
additional, high-resolution soil physical parameters may have the potential to improve the modelled relationship (e.g, soil type,
infiltration capacity, permeability, hydraulic conductivity).

Another source of structural uncertainty is the simplified representation of LULC as only four forest types (no, broadleaved,
conifer or mixed forest). According to Crozier (2010), the impact of human activity on slope stability is believed to be equal
to or even higher than effects of climate change. Additionally, human disturbances did certainly contribute to causing some
of the observed landslides (Hornich and Adelwöhrer, 2010). The Styrian state government projects an overall increase of the
region's population until the year 2060 (by +2.5 %), while peripheral regions are expected to show a decrease (der Steier-
märkischen Landesregierung, 2020). Demographic and economic changes may alter the regional demand for land for urban
development affecting the landscape composition. However, since detailed information on landscape impacts due to human
disturbance and socio-economic pathways was not available, we had to assume changes due to such anthropogenic factors to
be constant throughout our analysis.

## 5.3   Model Assessment

In this study, we successfully modified a GAM for landslide susceptibility analysis to explicitly account for spatial autocorrela-
tion effects using a GP component (GAM-Spatial) - an often missed feature in landslide susceptibility modelling (Reichenbach
et al., 2018). The spatial GP component explained a meaningful fraction of the deviance (6 % mDD), indicating that a spa-
tial pattern remained unaccounted for by the other predictors in the model. Furthermore, the modified GAM showed similar
predictor-response relationships and variable importances as well as an outstanding discrimination skill (mAUROC of 0.94)
comparable to GAMs used in previous studies (Knevels et al., 2020; Maraun et al., 2022), confirming the general reliability of
the landslide susceptibility model.

For the assessment of the uncertainties related to GAM-Spatial, we identified and applied optimal hyperparameters for a
simple Metropolis Hastings sampler (*mgcv::gam.mh* function in R) to stochastically simulate GAMs by approximating the
posterior distribution of the coefficients, as this approach is recommended for logit link functions (Wood, 2015). However,
the identified very large optimal hyperparameter value for *t.df* (i.e. 6050 degrees of freedom of the *t* distribution) indicated
a nearly Gaussian approximation for the posterior distribution. Furthermore, regarding the similarity in the critical values for
simultaneous intervals between the Metropolis Hastings sampler and a simple Gaussian approximation (*mgcv::rmvn* function




in R, Table A7 in Appendix A), the effort put into tuning and applying the Metropolis Hastings sampler is worth reconsidering
in future studies with similar settings.

## 5.4 Applicability

The presented methodological approaches are generally applicable to other regions. However, the established landslide suscep-
tibility model was based on only two landslide-event inventories at the regional scale for a single study area. Thus, the projected
uncertainties and effects of climate and LULC change on rainfall-induced landslide occurrences may be specific to the local
geological and geomorphological setting. With a greater awareness of the need for a reliable landslide-event inventory and with
additional landslides surveyed in the future, a more generalizable and less uncertain landslide model may be established.

As a practical application, the prediction of landslide susceptibility, its projections of change in area and magnitude with
climate change, and its uncertainties may provide important tools for planners and decision-makers. With the chosen approach
it is possible to consider susceptibility maps not only statically and retrospectively, but also to project them into the future,
which is an essential prerequisite for climate change adaptation in the context of slope stability. In the spatial context, recom-
mendations for the construction of new settlement infrastructure may be derived based on a map, and zones of high landslide
susceptibility along with uncertainty graphs can be communicated to local planners and environmental managers (cf. Petschko
et al., 2014). In the engineering context, grey (e.g., engineering structure) or green approaches (e.g., forest management) may
be considered as a scenario, and their effectiveness for potential slope stabilisation on a regional scale may be assessed (cf. De-
bele et al., 2019). Such development scenarios also enable the analysis of the exposure of elements at risk, potential mitigation
measurements and sustainable adaptations plans.

## 6 Conclusions

In this study, we analysed uncertainties in storylines of high landslide susceptibility in a changing environment. We established
a landslide model based on two large rainfall-triggered landslide events in the Styrian basin (2009 and 2014), showing the
direct link of hydrometeorological and LULC pattern to landslide occurrences while also accounting for residual spatial auto-
correlation. We identified distinct signals of projected changes in highly susceptible areas depending on the storyline. In the
worst-case of 4 K warming, *much heavier rain* may cause an increase of 35 % in highly susceptible area with additionally 37 %
higher chances of landslide occurrence (HadGEM), while *much drier soils* might even over-compensate this effect, leading to
more stable slopes (IPSL in 3 K warming, OR of 0.6 and −45 % in high susceptible landslide area). Proactive land-use man-
agement (i.e., afforestation and climate-resilient forest conversion) has the potential to reduce the extent of highly susceptible
areas. In the NO-CC scenario (i.e. pre-industrial, counterfactual climate), the highly susceptibility was generally estimated to
be lower (exception: GFDL).

The assessed landslide susceptibility uncertainty accounted for climate change and parametric landslide model uncertainty,
and its uncertainty cascade was generally high. Even though for some climate models and hydrometeorological variables,
within-event internal climate model variability showed opposed delta change factors (e.g., GFDL for five-day rainfall in



PARIS), their associated uncertainties in the landslide predictions were by far lower than uncertainties arising from the land-slide model (ratio of around 0.25). Furthermore, parametric landslide model uncertainty was found to be of the same order as the climate scenario uncertainty in the higher warming level (+3 K and +4 K). In particular, the most important predictor-response relationship, the five-day rainfall, was identified to be the most uncertain not only for the landslide susceptibility

model, but also in the within-event internal climate model variability assessment. For studies focussing on the climate signal or forced change, within-event internal climate model variability can be accounted for by averaging across the ensemble members (Maraun et al., 2022).

The compound characteristic of the extreme weather event in the Styrian basin in 2009 made an integrated modelling frame-work necessary. The assessment of future landslide occurrences and their associated uncertainties considering environmental

change is a challenging task due to its complexity, feedbacks, and the requirement to involve multiple disciplines and their associated methodological uncertainties. However, the analysis of compound events and their uncertainties is crucial to under-standing and improving the underlying key processes, and thus needed to support local decision makers and spatial planners in risk assessment (Zscheischler et al., 2020; Kirchner et al., 2021). With a higher awareness of the local institutes for the creation of reliable landslide inventories and the support of high-resolution ground data, model uncertainties may be reduced.

*Code and data availability.* The calibrated landslide susceptibility models can be accessed from https://doi.org/10.5281/zenodo.6365228. Landslide data can be requested from the State of Styria (raimund.adelwoehrer@stmk.gv.at), Geological Survey (GBA, arben.kociu@geolba.ac.at), and the Institute of Military Geoinformation (IMG, helene.kautz@bmlv.gv.at). Rainfall data from the Integrated Nowcasting through Com-prehensive Analysis (INCA) System is available from the Austrian Meteorological Service (https://data.hub.zamg.ac.at). Climate model data for the IFS boundary conditions for the CCLM RCM can be obtained from the ECMWF (https://www.ecmwf.int/en/forecasts/datasets(cycle35r2)),

ERA5 and ERA-Interim reanalysis data from ECMWF are available from https://www.ecmwf.int/en/forecasts/ datasets/browse-reanalysis-datasets, and data from the chosen CMIP5 GCM simulations can be downloaded from the Climate and Environmental Retrieval and Archive (CERA) Database (https://cera-www.dkrz.de/WDCC/ui/cerasearch).

**Appendix A**



**Table A1.** Overview of landslide susceptibility models

| GAM | Variables | Reference |
|---|---|---|
| GAM-Co | land surface variables, meteorological variables, geology, LULC | Knevels et al. (2020) |
| GAM-SM | GAM-Co, soil moisture | Maraun et al. (2022) |
| GAM-SM+TC | GAM-SM, five-day rainfall 80 mm top-coded | Maraun et al. (2022) |
| GAM-Spatial | GAM-SM+TC, Gaussian process smoother | this study |

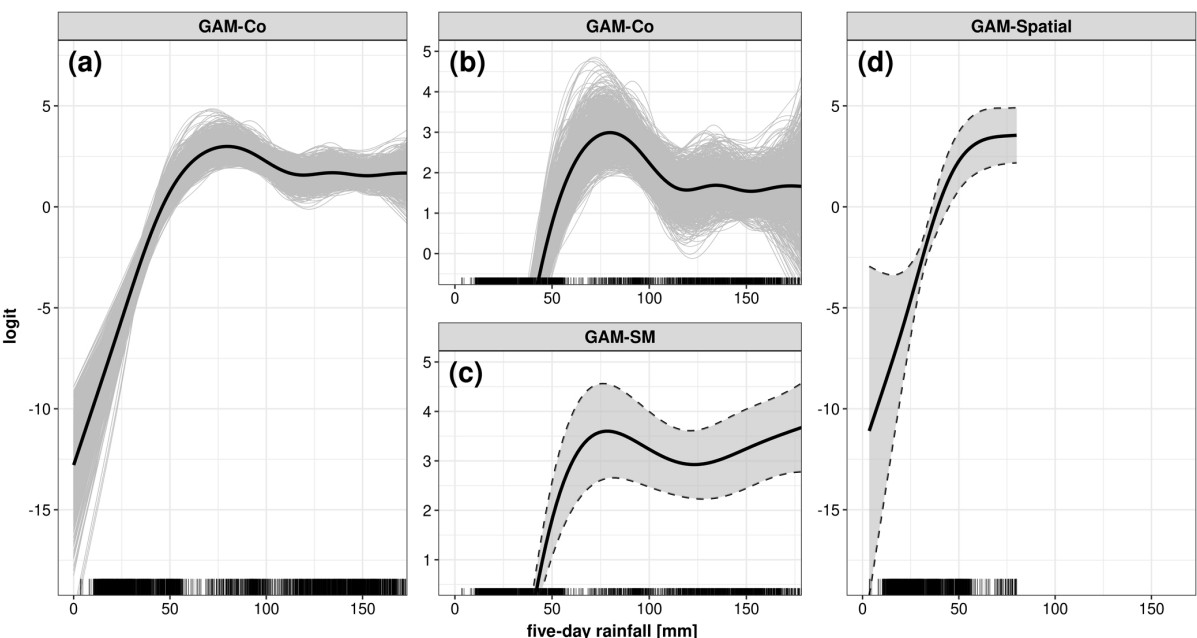

**Figure A1.** Response-predictor relationships of five-day rainfall. (a) GAM-Co from Knevels et al. (2020) and (b) enlarged. (c) GAM-SM from Maraun et al. (2022) (enlarged). (d) GAM-Spatial from this study. Note: 95 % pointwise Bayesian credible intervals are plotted for (c) and (d)





**Table A2.** Hydrometeorological storylines for the projected future climate

| Climate model | 5-day rainfall | Maximum 3-hours rainfall intensity | Soil moisture | Description |
|---|---|---|---|---|
| HadGEM2-CC | ++ | ++ | ∼ | much heavier rain |
| MIROC-ESM | ++ | ++ | − | much heavier rain, drier soil |
| IPSL-CM5A-MR | + | ++ | −− | heavier rain, much drier soil |
| GFDL-ESM2m | ∼ | ∼ | − | drier soil |

++ Strong increase; + increase; ∼ no change; − decrease; −− strong decrease. Adopted from Maraun et al. (2022).





**Table A3.** Delta change factors of hydrometeorological variables

| Climate Model | NO-CC | PARIS | 3 K | 4 K |
|---|---|---|---|---|
| *five-day rainfall* | | | | |
| GFDL-ESM2m | 0.99 [0.93; 1.06] $\perp$ | 1.01 [0.93; 1.08] $\perp$ | 1.01 [0.92; 1.09] $\perp$ | 1.00 [0.89; 1.07] $< \perp$ |
| HadGEM2-CC | 0.94 [0.89; 1.00] $\perp$ | 1.03 [0.95; 1.11] $\perp$ | 1.21 [1.07; 1.31] | 1.28 [1.11; 1.39] |
| IPSL-CM5A-MR | 0.97 [0.88; 1.04] $\perp$ | 1.03 [0.97; 1.10] $\perp$ | 1.08 [1.04; 1.16] | 1.10 [1.06; 1.18] |
| MIROC-ESM | 0.95 [0.86; 1.02] $\perp$ | 1.04 [0.98; 1.12] $\perp$ | 1.24 [1.19; 1.33] | 1.34 [1.29; 1.45] |
| *maximum three-hour rainfall intensity* | | | | |
| GFDL-ESM2m | 0.99 [0.92; 1.04] $\perp$ | 1.00 [0.94; 1.07] $> \perp$ | 1.00 [0.85;1.08] $< \perp$ | 0.99 [0.94; 1.06] $\perp$ |
| HadGEM2-CC | 0.90 [0.85; 0.96] | 1.04 [0.97; 1.13] $\perp$ | 1.31 [1.19; 1.41] | 1.42 [1.29; 1.55] |
| IPSL-CM5A-MR | 0.87 [0.82; 0.93] | 1.06 [1.00; 1.13] $\perp$ | 1.32 [1.23; 1.41] | 1.44 [1.34; 1.55] |
| MIROC-ESM | 0.87 [0.81; 0.93] | 1.08 [1.02; 1.14] | 1.39 [1.28; 1.51] | 1.61 [1.47; 1.75] |
| *soil moisture* | | | | |
| GFDL-ESM2m | 1.01 [1.01; 1.01] | 0.99 [0.99; 0.99] | 0.94 [0.94; 0.94] | 0.95 [0.94; 0.95] |
| HadGEM2-CC | 1.00 [1.00; 1.00] $<$ | 1.00 [1.00; 1.00] $>$ | 0.99 [0.99; 0.99] | 1.00 [1.00; 1.00] $>$ |
| IPSL-CM5A-MR | 1.02 [1.01; 1.02] | 0.98 [0.98; 0.99] | 0.84 [0.84; 0.84] | 0.89 [0.89; 0.89] |
| MIROC-ESM | 1.01 [1.01; 1.01] | 0.99 [0.99; 0.99] | 0.96 [0.96; 0.96] | 0.94 [0.94; 0.94] |

Mean is greater ($>$) or lower ($<$) than 1; $\perp$: 2.5th or 97.5th percentile showed contrasting delta change factors.




**Table A4.** Effective ranges of correlation functions

| Correlation function | REML Score | Rho | Effective Range |
|:---:|:---:|:---:|:---:|
| Sph | 3441.01 | 717 | 581.77 |
| Exp | 3444.83 | 237 | 709.99 |
| Gau | 3438.3 | 303 | 524.44 |
| Mat1 | 3438.99 | 126 | 597.73 |
| Mat2 | 3438.32 | 103 | 609.62 |
| Mat3 | 3438.59 | 88 | 605.15 |

Effective range was estimated using the R function *StempCens::EffectiveRange* for a
correlation $<= 0.05$.



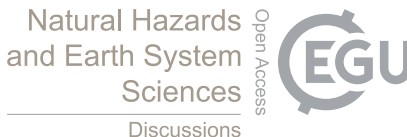

**Table A5.** Performance assessment (folds-based)

| Model | $\tilde{x}$ | $\overline{x}$ | $Range$ | $Min$ | $Max$ | $IQR$ |
|---|---|---|---|---|---|---|
| GAM-Spatial | 0.94 | 0.93 | 0.26 | 0.73 | 1 | 0.04 |
| GAM-SM+TC | 0.94 | 0.94 | 0.15 | 0.85 | 1 | 0.02 |
| GAM-Co | 0.94 | 0.94 | 0.15 | 0.84 | 0.99 | 0.02 |

Statistic: median ($\tilde{x}$), mean ($\overline{x}$), range ($Range$), minimum ($Min$), maximum ($Max$),
interquartile range ($IQR$).



**Table A6.** Variable importance

| Variable | GAM-Spatial | GAM-SM+TC | GAM-Co |
|---|---|---|---|
| *land surface variables* | | | |
| convergence index, 100 m | 0.12 (12) | 0.29 (11) | 0.34 (10) |
| convergence index, 500 m | 0.00 (18)[x] | 0.00 (16)[x] | 0.30 (11) |
| curvature, plan | 0.00 (15)[x] | 0.00 (14)[x] | 0.21 (14) |
| curvature, profile | 1.09 (5) | 1.40 (4) | 1.31 (4) |
| flow accumulation, logarithmized | 0.01 (13)[x] | 0.01 (12)[x] | 0.27 (13) |
| normalised height | 0.56 (10) | 0.70 (8) | 0.55 (9) |
| slope angle | 5.48 (4) | 7.09 (3) | 5.31 (3) |
| slope angle, catchment area | 0.00 (17)[x] | 0.00 (15)[x] | 0.08 (16) |
| slope aspect, S-N | 0.00 (14)[x] | 0.00 (17)[x] | 0.09 (15) |
| slope aspect, W-E | 0.00 (16)[x] | 0.00 (13)[x] | 0.27 (12) |
| TPI | 0.33 (11) | 0.46 (10) | 0.66 (8) |
| SWI | 0.76 (8) | 0.85 (7) | 0.71 (7) |
| *hydrometeorological variables* | | | |
| five-day rainfall | 10.76 (1) | 12.74 (1) | 15.13 (1) |
| maximum three-hour rainfall intensity | 0.85 (6) | 1.25 (5) | 1.05 (5) |
| soil moisture | 0.6 (9) | 0.57 (9) | |
| *geology* | | | |
| geology | 0.82 (7) | 1.06 (6) | 0.73 (6) |
| *land use/land cover* | | | |
| forest type | 8.8 (2) | 10.73 (2) | 8.57 (2) |
| *Gaussian process smooth* | | | |
| longitude, latitude | 5.98 (3) | | |

Variable importance measured in mean decrease in deviance explained (%), rank of variable in parentheses. Note:
Estimates for GAM-Co were based on SpCV models in Knevels et al. (2020). [x] model term was shrunk to zero.

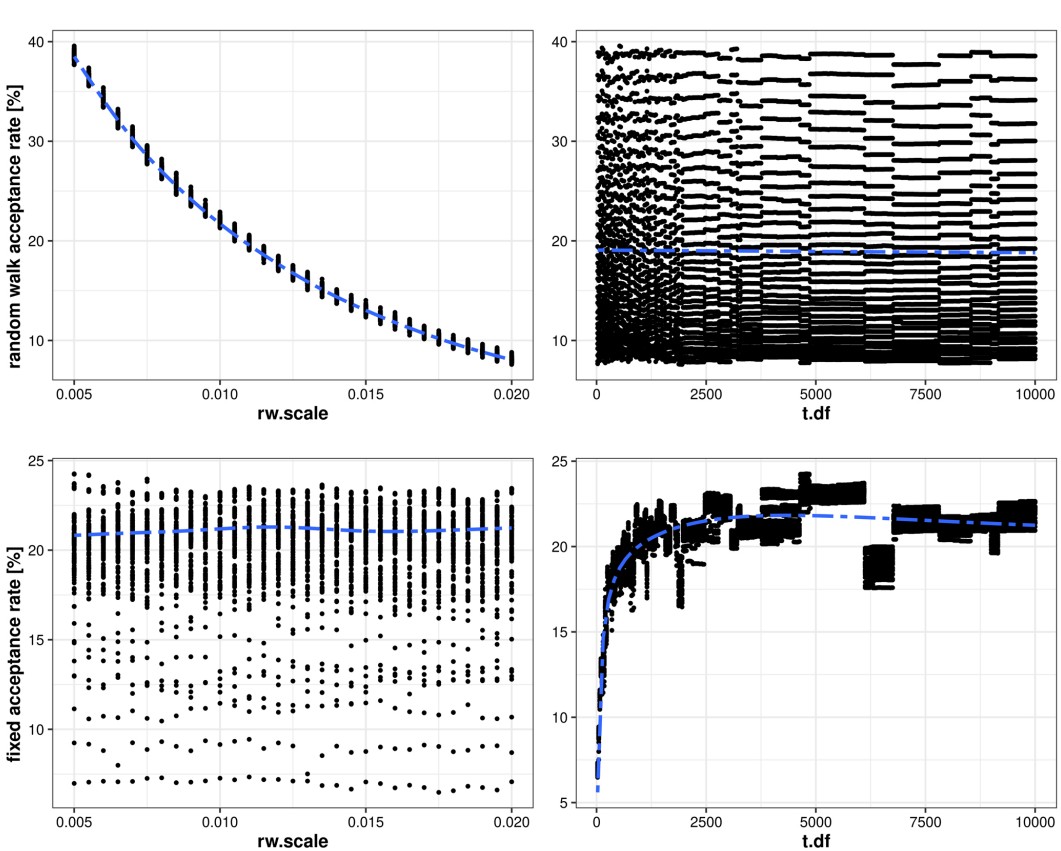

**Figure A2.** Relationship of *mgcv::gam.mh*'s hyperparameter values to acceptance rates. Blue dashed line: Smooth function.



**Table A7.** Critical values for simultaneous credible intervals of non-parametric transformation functions at the 95 % credible level

| Variable | GAM-Spatial | | GAM-SM |
| --- | --- | --- | --- |
| | *gam.mh* | *rmvn* | *rmvn* |
| *land surface variables* | | | |
| convergence index, 100 m | 1.982 (1.01) | 1.922 (0.98) | 3.178 (1.62) |
| convergence index, 500 m | 0.008[x] (0.00) | 0.008[x] (0.00) | 0.017[x] (0.00) |
| curvature, plan | 0.030[x] (0.02) | 0.028[x] (0.01) | 0.075[x] (0.04) |
| curvature, profile | 4.348 (2.22) | 4.120 (2.10) | 5.419 (2.77) |
| flow accumulation, logarithmized | 1.102[x] (0.56) | 1.082[x] (0.55) | 1.825[x] (0.93) |
| normalised height | 2.046 (1.04) | 1.987 (1.01) | 3.171 (1.62) |
| slope angle | 7.104 (3.62) | 6.724 (3.43) | 9.542 (4.87) |
| slope angle, catchment area | 0.021[x] (0.01) | 0.021[x] (0.01) | 0.062[x] (0.03) |
| slope aspect, S-N | 0.005[x] (0.00) | 0.005[x] (0.00) | 0.010[x] (0.01) |
| slope aspect, W-E | 0.002[x] (0.00) | 0.002[x] (0.00) | 0.005[x] (0.00) |
| TPI | 4.089 (2.09) | 3.848 (1.96) | 5.168 (2.64) |
| SWI | 3.081 (1.57) | 2.979 (1.52) | 3.325 (1.70) |
| *hydrometeorological variables* | | | |
| five-day rainfall | 37.846 (19.31) | 35.422 (18.07) | 22.650 (11.56) |
| maximum three-hour rainfall intensity | 2.149 (1.10) | 2.068 (1.05) | 3.003 (1.53) |
| soil moisture | 2.451 (1.25) | 2.388 (1.22) | 2.439 (1.24) |
| *Gaussian process smooth* | | | |
| longitude, latitude | 3.691 (1.88) | 3.616 (1.84) | |

Critical values are estimated at the 95 % credible level; *mgcv::gam.mh*: Simple posterior simulation with gam fits;
*mgcv::rmvn*: Generate from or evaluate multivariate normal or *t* densities. Note: Default coverage factor is 1.96 for 95 %
credible level. ()-brackets show the ratio to default. [x] model term was shrunk to zero. GAM-SM: GAM-Co including
soil-moisture, but no top-coded five-day rainfall.





**Table A8.** Uncertainty of predicted highly landslide susceptible area

| Scenario | GFDL | GFDL LULC | HadGem | HadGEM LULC | IPSL | IPSL LULC | MIROC | MIROC LULC |
|---|---|---|---|---|---|---|---|---|
| NO-CC | 5.1 | | 4.4 | | 4.8 | | 4.7 | |
| | [3.8; 6.8] | | [3.2; 6.1] | | [3.6; 6.4] | | [3.5; 6.3] | |
| PARIS | 4.7 | 4.3 | 5.1 | 4.6 | 4.7 | 4.3 | 5.0 | 4.5 |
| | [3.4; 6.5] | [3.1; 5.9] | [3.7; 6.9] | [3.4; 6.3] | [3.4; 6.6] | [3.1; 6.0] | [3.6; 6.8] | [3.3; 6.2] |
| 3 K | 3.6 | 3.3 | 5.9 | 5.4 | 2.7 | 2.5 | 5.4 | 4.9 |
| | [2.1; 5.4] | [1.9; 4.9] | [4.1; 8.2] | [3.8; 7.6] | [1.3; 5.0] | [1.1; 4.6] | [3.5; 8.0] | [3.2; 7.4] |
| 4 K | 3.7 | 3.3 | 6.6 | 6.1 | 4.0 | 3.7 | 5.7 | 5.3 |
| | [2.4; 5.4] | [2.2; 4.9] | [4.6; 9.3] | [4.2; 8.7] | [2.2; 6.7] | [2.0; 6.2] | [3.5; 8.9] | [3.2; 8.4] |

Note: Area relative to total study area. 95 % CI are based on within-event internal climate model variability and parametric landslide model uncertainty. For results for low, medium and high landslide susceptibility, please refer to Table S1 in the Supplementary Material.



**Table A9.** Odds ratios of landslide occurrences of high susceptibility class relative to present-day high landslide susceptibility

| Scenario | GFDL | GFDL LULC | HadGem | HadGEM LULC | IPSL | IPSL LULC | MIROC | MIROC LULC |
|---|---|---|---|---|---|---|---|---|
| NO-CC | 1.05 | | 0.91 | | 0.98 | | 0.96 | |
| | [1.00; 1.10] | | [0.87; 0.96] | | [0.93; 1.03] | | [0.92; 1.01] | |
| PARIS | 0.97 | 0.95 | 1.04 | 1.03 | 0.97 | 0.96 | 1.02 | 1.01 |
| | [0.92; 1.02] | [0.91; 1.01] | [0.98; 1.11] | [0.96; 1.09] | [0.92; 1.03] | [0.91; 1.02] | [0.97; 1.07] | [0.96; 1.06] |
| 3 K | 0.75 | 0.74 | 1.22 | 1.20 | 0.60 | 0.60 | 1.11 | 1.10 |
| | [0.67; 0.8] | [0.66; 0.79] | [1.12; 1.32] | [1.1; 1.3] | [0.55; 0.65] | [0.55; 0;65] | [1.02; 1.22] | [1.01; 1.21] |
| 4 K | 0.77 | 0.76 | 1.37 | 1.35 | 0.85 | 0.85 | 1.20 | 1.19 |
| | [0.73; 0.81] | [0.72; 0.8] | [1.24; 1.52] | [1.23; 1.5] | [0.78; 0.93] | [0.78; 0.93] | [1.08; 1.33] | [1.08; 1.32] |

Note: 95 % CI are based on within-event internal climate model variability. For results for low, medium and high landslide susceptibility, please refer to Table S2 in the Supplementary Material.





**Table A10.** Ratio of uncertainty sources in predicted high landslide susceptibility

| Scenario | GFDL | GFDL LULC | HadGem | HadGEM LULC | IPSL | IPSL LULC | MIROC | MIROC LULC |
|---|---|---|---|---|---|---|---|---|
| *Within-event intern climate variability to landslide model uncertainty \| Climate scenario to landslide model uncertainty* | | | | | | | | |
| NO-CC | 0.20 \| 0.28 | | 0.18 \| 0.28 | | 0.20 \| 0.29 | | 0.22 \| 0.29 | |
| PARIS | 0.19 \| 0.30 | 0.20 \| 0.34 | 0.24 \| 0.30 | 0.25 \| 0.33 | 0.19 \| 0.29 | 0.20 \| 0.32 | 0.18 \| 0.29 | 0.18 \| 0.31 |
| 3 K | 0.25 \| 1.28 | 0.27 \| 1.46 | 0.29 \| 1.07 | 0.29 \| 1.15 | 0.13 \| 1.02 | 0.13 \| 1.11 | 0.26 \| 0.97 | 0.27 \| 1.05 |
| 4 K | 0.15 \| 1.23 | 0.16 \| 1.38 | 0.34 \| 0.92 | 0.35 \| 1.00 | 0.19 \| 0.86 | 0.20 \| 0.95 | 0.27 \| 0.77 | 0.28 \| 0.82 |

For results for low, medium and high landslide susceptibility, please refer to Table S3 in the Supplementary Material.





*Author contributions.* Conceptualization, R.K. and A.B.; data curation, R.K., H.P. (Herwig Proske), D.M, and A.M.; formal analysis, R.K.;
methodology, R.K. and A.B.; supervision, H.P. (Helene Petschko) and A.B.; writing—original draft, R.K.; writing—review & editing, R.K.,
A.B., P.L., D.M., A.M., H.P. (Herwig Proske) and H.P. (Helene Petschko). All authors have read and agreed to the published version of the
manuscript.

*Competing interests.* The authors declare no conflict of interest.

*Acknowledgements.* This research was conducted within the Eastern Alpine Slope Instabilities under Climate Change (EASICLIM) project
funded by the Austrian Climate Research Program (ACRP), grant number KR16AC0K13160.



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
