# Peer review of "Assessing uncertainties in landslide susceptibility predictions in a changing environment (Styrian Basin, Austria)"

_Natural Hazards and Earth System Sciences, 2022_

## Author Comment (AC1)

**Response to Reviewer Comment on nhess-2022-154**
**Anonymous Referee #1**

*The manuscript is a good piece of science addressing a relevant issue, i.e. the evaluation of the uncertainties in landslide susceptibility modeling, taking into account also climate and environmental changes. The manuscript is clear, written in fluent English, and well organized. The figures and tables are useful for presenting and discussing the results. The introduction is complete and useful for focusing on the topic. The method is well presented and the discussion is clear. However, I've found some issues that need to be addressed and explained before the manuscript can be reconsidered for publication.*

Dear reviewer,

The coauthors and I are thankful for giving us the opportunity to submit a revised draft of our manuscript titled "Assessing uncertainties in landslide susceptibility predictions in a changing environment (Styrian Basin, Austria)" to Natural Hazards and Earth System Sciences. We are grateful for your encouraging, critical and constructive comments on this manuscript. The comments were taken fully into account in the revision, and we strongly believe that the comments and suggestions have largely increased the scientific value of the revised manuscript.

For our reply and revision of the manuscript we numbered the comments given by the referee. The comments by the reviewer are presented in black color, whereas our reply is in blue color. Additionally, we tracked changes using the latexdiff package in LaTeX in the revised manuscript.

– **Point 1:** Although earlier work by some of the authors is recalled in several places, the whole manuscript is rather long, so I would suggest trying to shorten it by at least 10% of the current length.

**Response 1:** We are thankful for the suggestion to shorten the manuscript. We acknowledge that the manuscript exceeds the expected number of journal pages, but we find that the work is succinctly written. We gave as briefly as possible only relevant details for model construction and reproducibility of the analysis. At some points we refer the reader to the cited publications for further detail. However, following the suggestion, we will be in touch with the editor if a shortening of the manuscript is required.

– **Point 2:** You use the term "storyline approach". I can't grab what you mean with "storyline" and "storyline approach". It seems to me that this term is not common in landslide analyses. I would suggest adding some explanation.

**Response 2**: Maraun et al. (2022) was to our knowledge the first who introduced "storylines" in the context of climate-change related landslide analysis. Even though the concept of storylines is not "new" (since 2018, Shepherd et al. 2018),

the emphasizing of this approach was recently highly promoted by the IPCC (Doblas-Reyes et al., 2021). The definition of a "storyline" is given at the beginning of chapter 2.2.2, however, we agree that a short explanation in the introduction may improve the understanding of the terminology. In the revised manuscript, we added in brackets (i.e. physically self-consistent, plausible pathways, Shepherd et al. 2018).

**Proposed change 2:**

Lines 31-33: Focusing on the rainfall event in 2009, Maraun et al. (2022) analysed the effect of projected future climate (2070–2100) and LULC changes on landslide occurrences using a storyline approach (i.e. physically self-consistent, plausible pathways, Shepherd et al. 2018) for the most-affected Feldbach region.

– **Point 3:** You defined the events that occurred in June 2009 and September 2014 as "extreme". How can you classify such events as "extreme"? Was a statistical analysis carried out?

**Response 3:** The reviewer correctly pointed out a possible source of confusion. For the 2009 event, Haiden (2009, 4) analyzed the amount of rainfall in less than 24 h in Styria, and grouped that rainfall event into events with a 50-year return period. Such an analysis was, however, not conducted for the 2014 event (nevertheless, as consequence, flood events corresponding to HQ50 and HQ100 were recorded at some places, see http://app.hydrographie.steiermark.at/berichte/september2014.pdf). Therefore, in the revised manuscript, we used the word "extreme" only in the context of the 2009 event, and "heavy" when accounted for both events.

**Proposed change 3:**

Lines 23-24: In June 2009 and September 2014, weather phenomena developed through a cut-off low brought heavy rainfall into the Styrian Basin, Austria (e.g., over 100 mm in 24 h in 2009).

Lines 26-27: The combined effect of premoisturing over the preceding winter and spring, and the actual triggering rainfall made these weather events into compound events [...].

Line 76: Study Area and  Rainfall Events

Line 85: In June 2009 and September 2014, heavy rainfall events occurred in southeast Styria.

Caption Figure 1: (b) Occurred landslides during  rainfall events.

Line 114: As landslide data, we used landslides that occurred during the heavy rainfall events, which were initially mapped by [...].

– **Point 4:** You added in the susceptibility analysis the rainfall data on the landslide failure day, i.e. the triggering precipitation conditions. I think this is questionable and in contrast with the theoretical definition of susceptibility (see e.g. Reichenbach et al. (2018) [https://doi.org/10.1016/j.earscirev.2018.03.001]; van Westen et al. (2008) [https://doi.org/10.1016/j.enggeo.2008.03.010]). Landslide susceptibility is "the likelihood of a landslide occurring in an area on the basis of the local terrain and environmental conditions", therefore the triggering rainfall conditions should

be removed from this analysis. You also wrote "For the landslide susceptibility analysis, we linked predisposing and triggering factors to landslide occurrences.". I think this can be considered a methodological issue.

**Response 4:** We agree with the reviewer that the term "landslide susceptibility" is still commonly understood as the likelihood of a certain area to be affected by landslide occurrence on the basis of local terrain conditions, which are assumed to be purely spatial ("where") and time-invariant. Already, Meusburger and Alewell (2009) questioned the validity of static landslide susceptibility maps under changing environmental conditions. But recently, this issue gained more attention (e.g. Jones et al., 2021; Ozturk et al., 2021), with several authors showing that the concept of time invariance is often violated on the time scale of few to several decades (e.g., Reichenbach et al. (2014) analyzing anthropogenic land use changes, Samia et al. (2017) analyzing "follow-up" landslides).

Therefore, we followed the recommendation of Gariano and Guzzetti (2016) and Reichenbach et al. (2018) to construct new models considering and investigating changes of environmental variables for landslide susceptibility (Gariano and Guzzetti 2016, 246: "We recommend to construct new slope stability models capable of cope with nonstationary climate and landslide records, and of considering the time dependence of the events.", Reichenbach et al. 2018, 84 "We recommend that studies should include climate-related variables in landslide susceptibility models.").

In our approach, local terrain conditions (e.g., predisposing factors such as slope angle, slope aspect, etc.), which are assumed not to change substantially in the course of centuries, are still considered time-invariant, but are extended by time-varying predictor variables (e.g., preparatory and triggering factors such as precipitation or land-use and land-cover). For clarification we enhanced the understanding of "landslide susceptibility" with additional information in the Landslide Susceptibility Model section.

**Proposed change 4:**

Lines 179-181: For the landslide susceptibility analysis, we linked predisposing and time-varying preparatory and triggering factors to landslide occurrences, by following recent recommendations for non-stationary landslide susceptibility models (Gariano and Guzzetti, 2016; Reichenbach et al., 2018; Jones et al., 2021; Ozturk et al., 2021).

Lines 514-516: Additionally, the time-varying modeling-perspective on landslide susceptibility as recommended by various authors (Gariano and Guzzetti, 2016; Reichenbach et al., 2018; Jones et al., 2021; Ozturk et al., 2021), allowed us to analyze the effects of LULC and climate change dynamics.

– **Point 5:** Regarding the environmental change simulation, you wrote (line 153) that "Adopting active forest management in the developed future LULC scenario, coniferous forest was replaced by climate resilient mixed forest".

**Response 5**: The authors agree with the referee that an in-depth analysis of effects of all possible kinds of land-use and land-cover changes on landslide occurrences is highly desirable. Actually, Maraun et al. (2022) analyzed a negative, "idealized" scenario representing extreme deforestation, i.e. one where all forest is removed (note: there was also a scenario with extreme afforestation). However, as the focus of the study is the assessment of uncertainties and not

primarily the effect of LULC changes, we decided to include only the "realistic" instead of the "idealized" scenario ("realistic" scenario was developed in close cooperation with the Regional Forestry Directorate and the District Forestry Authority). Furthermore, the LULC data used to fit our landslide model do not allow for further discrimination between different types of non-forested areas.

We added some explanations to refer the interested reader to Maraun et al. (2022) for these "idealized" scenarios.

**Proposed change 5:**

Lines 157-158: The developed, idealized scenarios of Maraun et al. (2022) were not in the scope of this analysis (i.e. extreme de- and afforestation).

– **Point 6:** Furthermore, you wrote (line 161) "Specifically, for each grid cell we determined the maximum three-hour rainfall intensity, and we took the maximum five-day rainfall." In my opinion, also this is questionable, given that it is not always the most severe rainfall condition during a meteorological event that can trigger landslides. An explanation is needed.

**Response 6:** We agree with the reviewer that there may be rainfall-triggering landslide events for which other precipitation aggregation schemes are more appropriate. Generally-spoken, there are many possibilities how to aggregate meteorological data, ranging from indices (e.g., landslide-rainfall index (Shou and Yang, 2015), antecedent rainfall index (Kirschbaum and Stanley, 2018)) to fixed moving sizes or number of days exceeding a certain amount of rainfall (e.g., in Gassner et al. 2015; Kim et al. 2015), and there is no general agreement on an optimal aggregation scheme. However, in Knevels et al. (2020, 9), we compared the two meteorological variables to "rainfall events responsible for landslides" extracted using the approach of Melillo et al. (2015). Specifically, for our landslides, we discovered a correlation of 0.95 between five-day rainfall and total amount during rainfall event, and of 0.58 between maximum three-hour rainfall intensity and amounts of rainfall sub-events; confirming the applicability of the presented aggregation scheme. To inform the interested reader where the rainfall aggregation comes from, we have now added an explanation in an appropriate place.

**Proposed change 6:**

Lines 105-107: Furthermore, precipitation was aggregated to obtain accumulated five-day rainfall (in mm) and maximum three-hour rainfall intensity (in mm h-1) on the landslide failure day, respectively (for the precipitation aggregation scheme, please refer to Knevels et al. 2020).

– **Point 7:** Finally, I suggest using round brackets for units of measurement. Please check all over the text.

**Response 7:** We are thankful for this hint. Regarding the submission-guideline for Mathematical notation and terminology (https://www.natural-hazards-and-earth-system-sciences.net/submission.html#math), it seems that there are no brackets around units in the fluid text. However, we replaced square-brackets with round brackets in the figures

**Proposed change 7:**

Figure 1: (mm); Figure 3: (mm), (mm h$^{-1}$), (%) + caption; Figure 4: (%); Figure 5a: (%); Figure A1: (mm); Figure A2: (%); Figure S1: (m); Figure S3: (mm), (mm h$^{-1}$), (%) + caption

**References**

Doblas-Reyes, F. J., Sörensson, A. A., Almazroui, M., Dosio, A., Gutowski, W. J., Haarsma, R., Hamdi, R., Hewitson, B., Kwon, W.-T., Lamptey, B. L., Maraun, D., Stephenson, T. S., Takayabu, I., Terray, L., Turner, A., and Zuo, Z.: Linking global to regional climate change, in: Climate Change 2021: The Physical Science Basis. Contribution of Working Group I to the Sixth Assessment Report of the Intergovernmental Panel on Climate Change, edited by Masson-Delmotte, V., Zhai, P., Pirani, A., Connors, S. L., Péan, C., Berger, S., Caud, N., Chen, Y., Goldfarb, L., Gomis, M. I., Huang, M., Leitzell, K., Lonnoy, E., Matthews, J. B. R., Maycock, T. K., Waterfield, T., Yelekçi, Ö., Yu, R., and Zhou, B., Cambridge University Press, 2021.

Gariano, S. L. and Guzzetti, F.: Landslides in a changing climate, Earth-Science Reviews, 162, 227–252, https://doi.org/10.1016/j.earscirev.2016.08.011, 2016.

Gassner, C., Promper, C., Beguería, S., and Glade, T.: Climate Change Impact for Spatial Landslide Susceptibility, in: Engineering Geology for Society and Territory, vol. 1, pp. 429–433, Springer International Publishing, https://doi.org/10.1007/978-3-319-09300-0_82, 2015.

Haiden, T.: Meteorologische Analyse des Niederschlags von 22.-25. Juni 2009 [Meteorological analysis of the precipitation from 22 to 25 June 2009], Tech. rep., ZAMG, Vienna, Austria, http://www.zamg.ac.at/docs/aktuell/2009-06-30_Meteorologische%20Analyse%20HOWA2009.pdf, last access: 20 May 2022, 2009.

Jones, J. N., Boulton, S. J., Bennett, G. L., Stokes, M., and Whitworth, M. R. Z.: Temporal Variations in Landslide Distributions Following Extreme Events: Implications for Landslide Susceptibility Modeling, Journal of Geophysical Research: Earth Surface, 126, e2021JF006 067, https://doi.org/10.1029/2021JF006067, 2021.

Kim, H. G., Lee, D. K., Park, C., Kil, S., Son, Y., and Park, J. H.: Evaluating Landslide Hazards Using RCP 4.5 and 8.5 Scenarios, Environmental Earth Sciences, 73, 1385–1400, https://doi.org/10.1007/s12665-014-3775-7, 2015.

Kirschbaum, D. and Stanley, T.: Satellite-Based Assessment of Rainfall-Triggered Landslide Hazard for Situational Awareness, Earth's Future, 6, 505–523, https://doi.org/10.1002/2017EF000715, 2018.

Knevels, R., Petschko, H., Proske, H., Leopold, P., Maraun, D., and Brenning, A.: Event-Based Landslide Modeling in the Styrian Basin, Austria: Accounting for Time-Varying Rainfall and Land Cover, Geosciences, 10, 217, https://doi.org/10.3390/geosciences10060217, number: 6 Publisher: Multidisciplinary Digital Publishing Institute, 2020.

Maraun, D., Knevels, R., Mishra, A. N., Truhetz, H., Bevacqua, E., Proske, H., Zappa, G., Brenning, A., Petschko, H., Schaffer, A., Leopold, P., and Puxley, B. L.: A severe landslide event in the Alpine foreland under possible future climate and land-use changes, Communications Earth & Environment, 3, 1–11, https://doi.org/10.1038/s43247-022-00408-7, number: 1 Publisher: Nature Publishing Group, 2022.

Melillo, M., Brunetti, M. T., Peruccacci, S., Gariano, S. L., and Guzzetti, F.: An Algorithm for the Objective Reconstruction of Rainfall Events Responsible for Landslides, Landslides, 12, 311–320, https://doi.org/10.1007/s10346-014-0471-3, 2015.

Meusburger, K. and Alewell, C.: On the Influence of Temporal Change on the Validity of Landslide Susceptibility Maps, Natural Hazards and Earth System Sciences, 9, 1495–1507, https://doi.org/10.5194/nhess-9-1495-2009, 2009.

Ozturk, U., Pittore, M., Behling, R., Roessner, S., Andreani, L., and Korup, O.: How Robust Are Landslide Susceptibility Estimates?, Landslides, 18, 681–695, https://doi.org/10.1007/s10346-020-01485-5, 2021.

Reichenbach, P., Busca, C., Mondini, A. C., and Rossi, M.: The Influence of Land Use Change on Landslide Susceptibility Zonation: The Briga Catchment Test Site (Messina, Italy), Environmental Management, 54, 1372–1384, https://doi.org/10.1007/s00267-014-0357-0, 2014.

Reichenbach, P., Rossi, M., Malamud, B. D., Mihir, M., and Guzzetti, F.: A review of statistically-based landslide susceptibility models, Earth-Science Reviews, 180, 60–91, https://doi.org/10.1016/j.earscirev.2018.03.001, 2018.

Samia, J., Temme, A., Bregt, A., Wallinga, J., Guzzetti, F., Ardizzone, F., and Rossi, M.: Do landslides follow landslides? Insights in path dependency from a multi-temporal landslide inventory, Landslides, 14, 547–558, https://doi.org/10.1007/s10346-016-0739-x, 2017.

Shepherd, T. G., Boyd, E., Calel, R. A., Chapman, S. C., Dessai, S., Dima-West, I. M., Fowler, H. J., James, R., Maraun, D., Martius, O., Senior, C. A., Sobel, A. H., Stainforth, D. A., Tett, S. F. B., Trenberth, K. E., van den Hurk, B. J. J. M., Watkins, N. W., Wilby, R. L., and Zenghelis, D. A.: Storylines: an alternative approach to representing uncertainty in physical aspects of climate change, Climatic Change, 151, 555–571, https://doi.org/10.1007/s10584-018-2317-9, 2018.

Shou, K.-J. and Yang, C.-M.: Predictive Analysis of Landslide Susceptibility under Climate Change Conditions — A Study on the Chingshui River Watershed of Taiwan, Engineering Geology, 192, 46–62, https://doi.org/10.1016/j.enggeo.2015.03.012, 2015.

---

## Author Comment (AC2)

**Response to Reviewer Comment on nhess-2022-154**
**Anonymous Referee #2**

*This paper investigated the uncertainty cascade in storylines of landslide susceptibility emerging from climate change and parametric landslide model uncertainty. In general, this paper is interesting and rich in content. However, there are many mistakes in the basic concept of landslide susceptibility, hazard and risk assessment. In terms of landslide susceptibility prediction modeling process, the writing of this paper is rather rough. In terms of organizational structure, the thesis is difficult to understand. Therefore, it is recommended to reject the paper.*

Dear reviewer,

The coauthors and I are thankful for the review. We regret that the topic was difficult to understand and that the methods were not clear enough. We try to elucidate the methodological approach for the reader with specific adaptations and to clarify the misunderstandings regarding the definition of susceptibility. We would be grateful for your constructive comments on this manuscript regarding critical points, which are at the current state not clear for us. Furthermore, we would be very pleased if a revised version of the manuscript is still under consideration for publication. The referee's comments were taken into account in the revision, and we strongly believe that the comments and suggestions have largely increased the scientific value of the revised manuscript.

For our reply and revision of the manuscript we numbered the comments given by the referee. The comments by the reviewer are presented in black color, whereas our reply is in blue color. Additionally, we tracked changes using the latexdiff package in LaTeX in the revised manuscript.

– **Point 1:** Landslide susceptibility refers to the spatial probability of landslide occurrence affected by landslides themselves conditioning factors, without considering triggering factors such as heavy rainfall, earthquake, et al. Landslide hazard refers to the spatial and time probability of landslide occurrence under condition factors and trigger factors. Hence, this paper focus on landslide susceptibility affected by land cover change and heavy rainfall. I believe this paper has problems with the basic concepts of landslide susceptibility.

**Response 1:** We agree with the reviewer that the term "landslide susceptibility" is still commonly understood as the likelihood of a certain area to be affected by landslide occurrence on the basis of local terrain conditions, which. are assumed to be purely spatial ("where") and time-invariant. The authors would like to emphasize that this is a purely terminological issue.

However, already, Meusburger and Alewell (2009, in NHESS) questioned the validity of static landslide susceptibility maps under changing environmental conditions. But recently, this issue gained more attention (e.g. Jones et al., 2021; Ozturk et al., 2021), with several authors showing that the concept of time invariance is often violated on the time scale of few to several decades (e.g., Reichenbach et al. (2014) analyzing anthropogenic land use changes, Samia et al. (2017) analyzing "follow-up" landslides). Therefore, we followed the recommendation of Gariano and Guzzetti (2016, 246), Reichenbach et al. (2018, 84) and other authors (Jones et al., 2021; Ozturk et al., 2021) to construct new models considering and investigating changes of environmental variables for landslide susceptibility.

In our approach, local terrain conditions (e.g., predisposing factors such as slope angle, slope aspect, etc.), which are assumed not to change substantially in the course of centuries, are still considered time-invariant, but are extended by time-varying predictor variables (e.g., preparatory and triggering factors such as soil moisture, precipitation or land-use and land-cover). Furthermore, in the entire manuscript the terms "hazard" and "risk" are only used in the context of damages or fatalities and in the broader context of management, which fits very well the in Brabb (1984) defined terminologies.

For clarification we enhanced the understanding of "landslide susceptibility" with additional information in the Landslide Susceptibility Model section.

**Proposed change 1:**

Lines 179-181: For the landslide susceptibility analysis, we linked predisposing and time-varying preparatory and triggering factors to landslide occurrences, by following recent recommendations for non-stationary landslide susceptibility models (Gariano and Guzzetti, 2016; Reichenbach et al., 2018; Jones et al., 2021; Ozturk et al., 2021).

Lines 514-516: Additionally, the time-varying modeling-perspective on landslide susceptibility as recommended by various authors (Gariano and Guzzetti, 2016; Reichenbach et al., 2018; Jones et al., 2021; Ozturk et al., 2021), allowed us to analyze the effects of LULC and climate change dynamics.

– **Point 2:** The writing ideas of this paper are very confused, and it is difficult for people to understand the specific steps and methods of the research. Especially in the introduction and methods section.

**Response 2**: We are thankful for the referee pointing out sources of confusion and misunderstanding. Regarding the methods, we have added a flow chart in the appendix showing a general overview of methodical approach from the beginning - model fitting, to the end - uncertainty ratios. Regarding the introduction, we more specifically describe the term storyline and emphasize more clearly our objectives.

**Proposed change 2:**

Lines 31-33: Focusing on the rainfall event in 2009, Maraun et al. (2022) analysed the effect of projected future climate (2070–2100) and LULC changes on landslide occurrences using a storyline approach (i.e. physically self-consistent, plausible pathways, Shepherd et al. 2018) for the most-affected Feldbach region.

Lines 66-67: The focus was not on creating landslide susceptibility maps, but on the quantification of uncertainties.

Lines 169-171: The proposed approach to assessing landslide susceptibility uncertainty addresses model fitting accounting for spatial dependency (Sect. 3.1), predictions considering environmental change (Sect. 3.2), and their amount of uncertainty (Sect. 3.3). Please refer to Fig. A1 in the Appendix A for an overview.

[Figure]

**Figure 1.** Overview of the proposed landslide susceptibility prediction and uncertainty assessment. (Note, the figure number does not match with the revised manuscript - Fig. A1 in the Appendix A)

– **Point 3:** Where are the input and output variables of landslide susceptibility prediction modeling described in this paper?

**Response 3:** The reviewer correctly pointed out a missing linkage to the Knevels et al. (2020) reference. The input variables were briefly mentioned at the second paragraph of the chapter in which the landslide model was introduced (3.1 Landslide Susceptibility Model, lines 173-177). In Knevels et al. (2020) further details are available on how these input variables were derived (software, hyperparameter settings etc.). As this paper is already quite extensive in pages, we shortened this description. In the revised manuscript we added the information that more details on the delineation can be found in Knevels et al. (2020).

**Proposed change 3:**

Lines 185-186: For more information on the delineation of the predictor variables and their maps, refer to Knevels et al. (2020) and Fig. S6 in the Supplementary Material, respectively.

– **Point 4:** How is the uncertainty problem concerned in this paper quantified?

**Response 4:** The authors thank the referee for the hint to a possible source of confusion. In last paragraph of the chapter "3.3 Uncertainty Cascade in Landslide Predictions" (lines 285 - 289), the authors described how the ratios are delineated: "Finally, to compare the different sources of uncertainty in the uncertainty cascade, we calculated the ratio of the uncertainty spread of within-event internal climate model variability and scenario uncertainty (i.e., spread of climate signals), respectively, to the parametric uncertainty spread of the landslide model. The joint uncertainty distribution of the landslide models' parametric uncertainty and within-event internal climate model variability is here referred to as the storyline uncertainty."

In the revised manuscript, we introduced a mathematical notation and changed the term "spread" to "width" to prevent the possibility of confusion. Furthermore, we hope that Change 2 (flow chart of methodological approach) improves the understanding of the analysis.

**Proposed change 4:**

Line 294-300: (Note, the equation numbers do not match with the revised manuscript)

Finally, to compare the different sources of uncertainty in the uncertainty cascade, we calculated the following ratios:

$$R_{IV;Lsl} = \frac{width \ of \ CI_{IV}}{width \ of \ CI_{Lsl}} \tag{1}$$

$$R_{CS;Lsl} = \frac{width \ of \ CI_{CS}}{width \ of \ CI_{Lsl}} \tag{2}$$

where $R_{IV;Lsl}$ and $R_{CS;Lsl}$ denote the ratio of the uncertainty width of within-event internal climate model variability ($CI_{IV}$) and scenario uncertainty (i.e., width of climate signals, $CI_{CS}$), respectively, to the parametric uncertainty width of the landslide model ($CI_{Lsl}$). The joint uncertainty distribution of the landslide models' parametric uncertainty and within-event internal climate model variability is here referred to as the storyline uncertainty ($CI_{Story}$).

– **Point 5:** Where are the environmental factor maps of landslide susceptibility and landslide susceptibility outcome maps? These problems [Points 3-5] are not well reflected.

**Response 5**: The reviewer pointed out a possible source of misunderstanding. This article is not about creating landslide maps, but to quantify uncertainties in predictions emerging from the landslide and climate models. However, we followed the reviewer and added two overview maps in the supplementary materials: a map showing all the input variables, and a map showing the landslide susceptibility classes of the event and selected climate model alongside with uncertainty graphs, which may be useful to communicate to local stakeholders.

Please note that the estimated predictor-response relationship, which were of interest in our analysis, were already made available via the supplementary materials (Figure S4 and S5, https://nhess.copernicus.org/preprints/nhess-2022-154/nhess-2022-154-supplement.pdf).

**Proposed change 5:**

Lines 184-185: For more information on the delineation of the predictor variables and their maps, refer to Knevels et al. (2020) and Fig. S6 in the Supplementary Material, respectively.

Lines 534-537: In the spatial context, recommendations for the construction of new settlement infrastructure may be derived based on a map, and zones of high landslide susceptibility along with uncertainty graphs (Fig. S7 in the Supplementary Material) can be communicated to local planners and environmental managers (cf. Petschko et al., 2014).

[Figure]

**Figure 2.** Overview of input variables for predicting landslide susceptibility of the 2009 event. Geology: 0: 'Others', 1: 'Neogene formations dominated by fine-grained sediments', 2: 'Neogene formations with coarse-grained layers', 3: 'pre-Würmian Pleistocene formations', 4: 'Würm and holocene sediments'. (Note, the figure number does not match with the revised manuscript - Fig. S6 in the Supplementary Material)

[Figure]

**Figure 3.** Example of landslide susceptibility maps of the 2009 event and associated uncertainties in highly susceptible areas. Pre-industrial and future storylines are based on the HadGEM climate model in the 4 K warming scenario. Note: Uncertainty graphs are based on the entire study area. (Note, the figure number does not match with the revised manuscript - Fig. S7 in the Supplementary Material)

- **Point 6:** The figures are not clear enough.

  **Response 6:** The authors collectively went through all figures individually, and agreed that, except for Figure 5, all figures are clear. The authors modified Figure 5 by adding the mathematical notations introduced with Proposed Change 4, and thus improved its understanding.

  **Proposed change 6:** see Figure 5 in the revised manuscript.

- **Point 7:** The references are not new enough and are not representative enough.

  **Response 7:** The authors are wondering on what basis this statement was made. Nevertheless, the authors checked all references again on redundancy or the possibility for "updating", and modified them when it suited.

  **Proposed change 7:**

  Removed: Hastie and Tibshirani 1986; Nychka 1988; Oberkampf et al. 2004; Refsgaard et al. 2007; Brenning 2012; Fressard et al. 2014; Hussin et al. 2016

  Added: Brock et al. 2020; Ozturk et al. 2021; Jones et al. 2021

- **Point 8:** The uncertainty characteristics are assessed by which indexes? These description are not clear.

  **Response 8:** Please refer to **Point 4**.

  **Proposed change 8:** see **Proposed Change 4**.

- **Point 9:** There is insufficient analysis of feasible solutions to the problems in this paper.

  **Response 9:** The problem addressed in this paper is the assessment of uncertainties - not the reduction of uncertainties. Nevertheless, the authors even gave some ideas, how these analyzed uncertainties might be reduced (e.g. Lines 440, 450-453 for the climate uncertainties or Lines 482-485, 515-516 for the landslide model uncertainties). The authors have shown that the chosen approach to uncertainty assessment is valid and (evidently) feasible. We are therefore confident that in reviewing these text sections, the referee will appreciate our contributions mainly to the assessment of uncertainties.

  **Proposed change 9:** No change made

**References**

Brabb, E. E.: Innovative Approaches to Landslide Hazard and Risk Mapping, in: Fourth International Symposium on Landslides, vol. 1, pp. 307–328, Canadian Geotechnical Society, 1984.

Brenning, A.: Improved spatial analysis and prediction of landslide susceptibility: Practical recommendations, in: Landslides and Engineered Slopes: Protecting Society through Improved Understanding, edited by Eberhardt, E., Froese, C., Turner, A. K., and Leroueil, S., vol. 1, pp. 789–794, CRC Press/Balkema, Banff, Canada, 2012.

Brock, J., Schratz, P., Petschko, H., Muenchow, J., Micu, M., and Brenning, A.: The Performance of Landslide Susceptibility Models Critically Depends on the Quality of Digital Elevations Models, Geomatics, Natural Hazards and Risk, 11, 1075–1092, https://doi.org/10.1080/19475705.2020.1776403, 2020.

Fressard, M., Thiery, Y., and Maquaire, O.: Which data for quantitative landslide susceptibility mapping at operational scale? Case study of the Pays d'Auge plateau hillslopes (Normandy, France), Natural Hazards and Earth System Sciences, 14, 569–588, https://doi.org/10.5194/nhess-14-569-2014, publisher: Copernicus GmbH, 2014.

Gariano, S. L. and Guzzetti, F.: Landslides in a changing climate, Earth-Science Reviews, 162, 227–252, https://doi.org/10.1016/j.earscirev.2016.08.011, 2016.

Hastie, T. and Tibshirani, R.: Generalized Additive Models, Statistical Science, 1, 297–310, https://doi.org/10.1214/ss/1177013604, 1986.

Hussin, H. Y., Zumpano, V., Reichenbach, P., Sterlacchini, S., Micu, M., van Westen, C., and Bălteanu, D.: Different landslide sampling strategies in a grid-based bi-variate statistical susceptibility model, Geomorphology, 253, 508–523, https://doi.org/10.1016/j.geomorph.2015.10.030, 2016.

Jones, J. N., Boulton, S. J., Bennett, G. L., Stokes, M., and Whitworth, M. R. Z.: Temporal Variations in Landslide Distributions Following Extreme Events: Implications for Landslide Susceptibility Modeling, Journal of Geophysical Research: Earth Surface, 126, e2021JF006 067, https://doi.org/10.1029/2021JF006067, 2021.

Knevels, R., Petschko, H., Proske, H., Leopold, P., Maraun, D., and Brenning, A.: Event-Based Landslide Modeling in the Styrian Basin, Austria: Accounting for Time-Varying Rainfall and Land Cover, Geosciences, 10, 217, https://doi.org/10.3390/geosciences10060217, number: 6 Publisher: Multidisciplinary Digital Publishing Institute, 2020.

Maraun, D., Knevels, R., Mishra, A. N., Truhetz, H., Bevacqua, E., Proske, H., Zappa, G., Brenning, A., Petschko, H., Schaffer, A., Leopold, P., and Puxley, B. L.: A severe landslide event in the Alpine foreland under possible future climate and land-use changes, Communications Earth & Environment, 3, 1–11, https://doi.org/10.1038/s43247-022-00408-7, number: 1 Publisher: Nature Publishing Group, 2022.

Meusburger, K. and Alewell, C.: On the Influence of Temporal Change on the Validity of Landslide Susceptibility Maps, Natural Hazards and Earth System Sciences, 9, 1495–1507, https://doi.org/10.5194/nhess-9-1495-2009, 2009.

Nychka, D.: Bayesian Confidence Intervals for Smoothing Splines, Journal of the American Statistical Association, 83, 1134–1143, https://doi.org/10.1080/01621459.1988.10478711, 1988.

Oberkampf, W. L., Helton, J. C., Joslyn, C. A., Wojtkiewicz, S. F., and Ferson, S.: Challenge problems: uncertainty in system response given uncertain parameters, Reliability Engineering & System Safety, 85, 11–19, https://doi.org/10.1016/j.ress.2004.03.002, 2004.

Ozturk, U., Pittore, M., Behling, R., Roessner, S., Andreani, L., and Korup, O.: How Robust Are Landslide Susceptibility Estimates?, Landslides, 18, 681–695, https://doi.org/10.1007/s10346-020-01485-5, 2021.

Petschko, H., Brenning, A., Bell, R., Goetz, J., and Glade, T.: Assessing the quality of landslide susceptibility maps – case study Lower Austria, Natural Hazards and Earth System Sciences, 14, 95–118, https://doi.org/10.5194/nhess-14-95-2014, 2014.

Refsgaard, J. C., van der Sluijs, J. P., Højberg, A. L., and Vanrolleghem, P. A.: Uncertainty in the environmental modelling process – A framework and guidance, Environmental Modelling & Software, 22, 1543–1556, https://doi.org/10.1016/j.envsoft.2007.02.004, 2007.

Reichenbach, P., Busca, C., Mondini, A. C., and Rossi, M.: The Influence of Land Use Change on Landslide Susceptibility Zonation: The Briga Catchment Test Site (Messina, Italy), Environmental Management, 54, 1372–1384, https://doi.org/10.1007/s00267-014-0357-0, 2014.

Reichenbach, P., Rossi, M., Malamud, B. D., Mihir, M., and Guzzetti, F.: A review of statistically-based landslide susceptibility models, Earth-Science Reviews, 180, 60–91, https://doi.org/10.1016/j.earscirev.2018.03.001, 2018.

Samia, J., Temme, A., Bregt, A., Wallinga, J., Guzzetti, F., Ardizzone, F., and Rossi, M.: Do landslides follow landslides? Insights in path dependency from a multi-temporal landslide inventory, Landslides, 14, 547–558, https://doi.org/10.1007/s10346-016-0739-x, 2017.

Shepherd, T. G., Boyd, E., Calel, R. A., Chapman, S. C., Dessai, S., Dima-West, I. M., Fowler, H. J., James, R., Maraun, D., Martius, O., Senior, C. A., Sobel, A. H., Stainforth, D. A., Tett, S. F. B., Trenberth, K. E., van den Hurk, B. J. J. M., Watkins, N. W., Wilby, R. L., and Zenghelis, D. A.: Storylines: an alternative approach to representing uncertainty in physical aspects of climate change, Climatic Change, 151, 555–571, https://doi.org/10.1007/s10584-018-2317-9, 2018.

---

## Author Response (AR1)

**Response to Editor and Reviewer Comments on nhess-2022-154**

Dear editor and reviewers,

The coauthors and I are thankful for giving us the opportunity to submit a revised draft of our manuscript titled "Assessing uncertainties in landslide susceptibility predictions in a changing environment (Styrian Basin, Austria)" to Natural Hazards and Earth System Sciences. We are grateful for your encouraging, critical and constructive comments on this manuscript. Furthermore, we would be very pleased if a revised version of the manuscript is still under consideration for publication. The comments were taken fully into account in the revision, and we strongly believe that the comments and suggestions have largely increased the scientific value of the revised manuscript.

For our reply and revision of the manuscript we numbered the comments given by the referees. The comments by the reviewers and editor are presented in black color, whereas our reply is in blue color. Additionally, we tracked changes using the latexdiff package in LaTeX in the revised manuscript.

**Editor**

*On the basis of the referees' reports, and the main points summarized below, the paper could be accepted for publication should you be willing to apply some major modifications. The topic of the manuscript is of large interest to the NH community, and the approach used is quite novel and potentially providing insights into something that is still debated and unsolved. However, there are a few important issues present in the manuscript as highlighted in the review reports, which you will find attached, plus the following, partially overlapping, points:*

– **Point 1:** classical susceptibility does not usually include time - it is intended to produce a statistical inference on the relative probability in space. As a consequence, susceptibility is usually based on historical inventories and tries to predict the occurrence of landslides over a time span of the same order of magnitude as the time span of the inventory itself (uncertainties and missing information considered). Instead, you propose here event-based landslide maps to train a statistical model, which seems to be valid only for similar triggering events. In this way, to the extent that your climate-change forecasts are correct, you are predicting the relative response of the landscape to this event type, only. This poses two main concerns:

**Response 1:** The authors agree that the model is mostly applicable for rainfall-events with similar causal mechanisms on which the model was initially fitted. With more landslide inventories available any anomalous landslide distributions that

may have occurred in periods impacted by extreme events might be averaged out. - If that is of interest in a specific study. However, in the framework of Knevels et al. (2020), Maraun et al. (2022) and this study, we were especially interested in the landslide occurrences triggered by extreme weather events in 2009 (and 2014) and possible future pathways (see **Editor Point 1-A**).

Furthermore, the authors agree with the editor that we have challenged the "purely spatial and time-invariant" concept of landslide susceptibility by taking a dynamic perspective on the term. The questioning of the validity of static landslide susceptibility maps, especially under environmental change is not new (Meusburger and Alewell, 2009), but the issue gained recently more attention (Reichenbach et al., 2014; Samia et al., 2017; Lombardo et al., 2020; Jones et al., 2021; Ozturk et al., 2021) . In this study, we followed the recommendation of Gariano and Guzzetti (2016, 246) and Reichenbach et al. (2018, 84) to construct new models considering and investigating changes of environmental variables for landslide susceptibility. However, even this recommendation was already formulated at the beginning of the landslide susceptibility mapping process (e.g. Newman et al. 1978, 2; Nilsen et al. 1979, 88). The more conceptual debate on the definition of "landslide susceptibility" in regard to environmental change (uniformitarianism vs catastrophism) is outside the scope of this article.

In this study, we propose to combine local terrain conditions (predisposing factors), which are assumed not to change substantially in the course of centuries, and time-varying predictor variables (preparatory and triggering factors such as precipitation or land-use and land-cover, LULC) to create time-varying or conditional landslide susceptibility. Thus, the landslide susceptibility is conditional on the considered rainfall event(s), soil moisture and LULC.

**Change 1:**

Lines 188-192: For the landslide susceptibility analysis, we linked predisposing and time-varying preparatory and triggering factors to landslide occurrences, by following recent recommendations for non-stationary landslide susceptibility models (Gariano and Guzzetti, 2016; Reichenbach et al., 2018; Jones et al., 2021; Ozturk et al., 2021). Therefore, we are analyzing conditional landslide susceptibility in the sense of a susceptibility that applies under given (time-varying) environmental conditions.

Lines 524-526: Additionally, the time-varying modeling perspective on (conditional) landslide susceptibility as recommended by various authors (Gariano and Guzzetti, 2016; Reichenbach et al., 2018; Jones et al., 2021; Ozturk et al., 2021) allowed us to analyze the effects of LULC and climate change dynamics.

**Point 1-A:** How can then your samples (being based only on a few rare, documented events) be statistically representative of all the future expected extreme rainfall events?

**Response 1-A:** The purpose of our study is not to be representative of all future expected extreme rainfall events. The authors instead apply an event storyline approach (Shepherd et al., 2018; Doblas-Reyes et al., 2021). Maraun et al. (2022) were to our knowledge the first who introduced "storylines" in the context of climate-change related landslide analysis. In this context, the extreme weather event is simulated as it happened (here 2009), and subsequently physically

self-consistent, plausible pathways of this event are computed for pre-industrial and future climate (the event storylines). Therefore, we are not investigating "all the future expected extreme rainfall-events" but rather a similar event manifesting differently in the future under varied warming scenarios. This rather points to the question "how would this event occur in a warmer/colder climate, and what would be the associated landslide susceptibility." As we fit our models on this event, our sample is appropriate.

Our perspective allows, apart from the attribution of anthropogenic climate change to an actual event, also narratives for the future. Storylines are thus important tools for strengthening decision-making and communicating potential outcomes to local stakeholder. The storyline approach was recently promoted by the IPCC (Doblas-Reyes et al., 2021).

We extend the introduction of the term storyline in chapter *2.2.2 Environmental Change Simulations* to make this point more clear.

**Change 1-A:** *For changes regarding the storyline approach, please refer to* **Referee #1 Point #2**.

**Point 1-B:** How can you support (statistically) the inference that the effect produced by the triggering event X in position Y is able to produce the same effect in position Z (where local conditions are different from site Y)? I am not saying that you model is flawed, as I do not have sufficient information to say so, or the opposite. But maybe you could try to better frame the limits and conceptual context in which you are working. The approach may well be new, but at the moment we can't say if it is truly appropriate for the usual purposes of susceptibility mapping, that is scenario building, land planning, or risk reduction

**Response 1-B:** We are assuming that (1) the relationships we are modeling are not spatially variable (beyond the spatial variation we model), and (2) that there is no relevant spatial structure in the model residuals because we are incorporating all relevant confounders. - These are assumptions that are of course common to all spatial analyses, including the ones conducted to develop the susceptibility maps referred to by the reviewer, although these assumptions are rarely, if ever, made explicit by the authors of such studies.

**Point 2:** The main topic, despite being of general interest, is far from new. There are several papers published in the last 10 years that propose multi-temporal susceptibility across different climate change scenarios. It would be good if you listed at least some of them, to better set the background as well as to be able to better explain why your approach is novel. There are several papers, as well, that propose model sensitivity in statistical analysis of landslide hazard, including the influence of the different mapping parameters, scales, resolutions, sampling methods and so on. Please also enrich the list of references as related to this aspect. I do not want to add specific suggestions, as it is quite easy to find them. This would also help in framing your own research and the gaps you meant to fill.

**Response 2:** The authors want to point out that the main topic (see title) - the main objectives of this study is about uncertainty analysis, especially parametric landslide model uncertainty and climate model uncertainty. Therefore, lines 40 - 69 introduce definitions and some important literature relating to uncertainty studies (i.e., inventory biases; sampling

strategies; data quality, resolution and scale issues; spatial autocorrelation, or model validation). Literature on multi-temporal landslide susceptibility accounting also for environmental change which is related to this study can mainly be found in the discussion part (*5.1 Landslide Susceptibility in a Changing Environment*). The authors would like to emphasize the study of Maraun et al. (2022), which focuses on projected future landslide occurrences. However, we added some background information on the concept of storylines, to better frame the analysis.

**Change 2:** *For changes regarding the storyline approach, please refer to **Referee #1 Point #2**.*

**Point 3:** As a side note, more extreme weather events of the same type are available in the study region, as far as I know. Did you try to include them in the analysis? Or did you try to use them as external validation of your model? (see also references cited below on the Austrian Alps).

**Response 3:** The authors had no more information on landslides available in the Styrian basin (or the Southeastern Foreland, respectively) besides the two landslide events from 2009 and 2014. We are arware of "near" events such as the 2005-event in Gasen-Haslau. However, these landslides occurred in the Fischbacher Alps - a (sub)mountainous region having with different characteristics. Some historical LiDAR-based landslide inventories may in the meanwhile exist (as LiDAR-derived HRDTMs are more frequent available), but with the well-known drawbacks of unknown landslide age, biases in forested areas (over-report), and unknown trigger(ing rainfall). As a consequence, the other historical or event-based events are not suitable for external validation.

**Point 4:** Please consider that in the Eastern Alps, recent extreme events such as the Vaia storm of 2018 have highlighted that rainfall may start triggering landslides not just during the main storm, which only creates some preconditions, so to speak. But, instead, after several weeks. You use 5-day and 3-hour accumulated rain in your study. How can this influence your results? How can your statistics account for this delayed-time triggering

**Response 4:** The proposed landslide model also accounted for soil moisture, which may come into play in the editor's described case. However, as our model was not trained on such landslide observations, a model transfer may not be suitable.

Generally speaking, there are many possibilities on how to aggregate meteorological data and there is no general agreement on an optimal aggregation scheme. In Knevels et al. (2020, 9), we identified the applied aggregation schemes as suitable by comparing the rainfall variables to "rainfall events responsible for landslides" extracted using the approach of Melillo et al. (2015). However, for other studies different window-sizes may be appropriate.

Additionally, the authors would like to point out that we do not intend to forecast landslides.

**Change 4:** *For changes and an even more detailed response, please refer to **Referee #1 Point #6**.*

**Point 5:** In lines 14-15 you report UNISDR and CRED data: Economic Losses, Poverty & Disasters (1998 - 2017) that cover a variety of disasters around the globe; I would suggest including one more study to the introduction, which was

solely based on landslides, such as e.g.: i) Froude, M. J. and Petley, D. N.: Global fatal landslide occurrence from 2004 to 2016, Nat. Hazards Earth Syst. Sci., 18, 2161–2181, https://doi.org/10.5194/nhess-18-2161-2018, 2018.

**Response 5:** The authors decided to reference Wallemacq et al. (2018) as this report accounted for a longer period of time and the database (EM-DAT) to access the information on landslides was easily accessible. Unfortunately, in the meanwhile, one has to register to access the statistics: https://emdat.be/emdat_db/. We followed the editor's suggestion by also referencing Froude and Petley (2018), and we adjusted the number of fatalities accordingly, which seemed to be underestimated in the EM-DAT database.

**Change 5:**

Lines 14-16: During the period from 1998 to 2017, landslides affected 4.8 million people causing more than 55,997 deaths and over 5.28 billion US$ total damages (Froude and Petley, 2018; Wallemacq et al., 2018).

**Point 6:** In lines 19-20, you might want to consider and include the recent works related to landslide susceptibility in the Austrian alps, such as: i) Lima, P., Steger, S., & Glade, T. (2021). Counteracting flawed landslide data in statistically based landslide susceptibility modelling for very large areas: a national-scale assessment for Austria. Landslides, 18(11), 3531-3546. ii) Moharrami, M., Naboureh, A., Gudiyangada Nachappa, T., Ghorbanzadeh, O., Guan, X., & Blaschke, T. (2020). National-scale landslide susceptibility mapping in aaustria using fuzzy best-worst multi-criteria decision-making. ISPRS International Journal of Geo-Information, 9(6), 393. iii) Gudiyangada Nachappa, T., Kienberger, S., Meena, S. R., Hölbling, D., & Blaschke, T. (2020). Comparison and validation of per-pixel and object-based approaches for landslide susceptibility mapping. Geomatics, Natural Hazards and Risk, 11(1), 572-600

**Response 6:** The authors thank the editor for pointing out a possible source of confusion, and the recommendations of case studies. The Styrian basin is not located in the Austrian Alps, but in the southeastern Alpine forelands (Lieb and Embleton-Hamann, 2022). With regard to the different relative relief, alpine landslide studies may be faced with other predisposing characteristics. We hope that the modifications in the first paragraph of the introduction will help to clarify this point.

**Change 6:**

Lines 18-20: Across the Austrian Alps and their forelands, a generally high proneness of landslides is observed, which are among the main natural hazards frequently causing damage to houses and infrastructure as well as casualties (Jaedicke et al., 2014; Lima et al., 2021).

Lines 24-25: In June 2009 and September 2014, weather conditions developed through a cut-off low brought heavy rainfall into the Styrian Basin, Austria's southeastern Alpine forelands (e.g., over 100 mm in 24 h in 2009).

**Point 7:** In lines 63-64, you could try to address the bias associated with downscaling the climate data by using an approach as the one here: i) Roberts, DR, Wood, WH, Marshall, SJ. Assessments of downscaled climate data with a

high-resolution weather station network reveal consistent but predictable bias. Int J Climatol. 2019; 39: 3091– 3103. https://doi.org/10.1002/joc.6005

**Response 7:** We thank the editor for the hint to address biases or uncertainties related to climate data in this paragraph, and added briefly general limitations regarding that issue.

**Change 7:**

Lines 66-69: Regarding climate change, landslide studies often suffer from common modelling limitation. Namely, the presence of large-scale circulation errors in global climate models (GCMs), using GCMs without downscaling and subsequent bias adjustment (Roberts et al., 2019), not resolving convection in standard regional climate models (RCMs), and ignoring factors that might be potentially relevant in a changing climate such as soil moisture (Maraun et al., 2017, 2022).

**Point 8:** You provide no maps of susceptibility, which could help in identifying the locations of impending occurrences in a spatial way. Could you please add some of the maps (supplementary material as well)?

**Response 8:** The authors are thankful and followed this suggestion.

**Change 8:** *For changes and more details, please refer to **Referee #2 Point #5**.*

**Anonymous Referee #1**

- **Point 1:** Although earlier work by some of the authors is recalled in several places, the whole manuscript is rather long, so I would suggest trying to shorten it by at least 10% of the current length.

  **Response 1:** We are thankful for the suggestion to shorten the manuscript. We acknowledge that the manuscript exceeds the expected number of journal pages, but we find that the work is succinctly written. We gave as briefly as possible only relevant details for model construction and reproducibility of the analysis. At some points we refer the reader to the cited publications for further detail. However, based on the overall referees comments, the information given so far appears to be necessary to understand and to follow the study.

- **Point 2:** You use the term "storyline approach". I can't grab what you mean with "storyline" and "storyline approach". It seems to me that this term is not common in landslide analyses. I would suggest adding some explanation.

  **Response 2**: Maraun et al. (2022) were to our knowledge the first who introduced "storylines" in the context of climate-change related landslide analysis. Even though the concept of storylines is not "new" (since 2018, Shepherd et al. 2018), the emphasizing of this approach was recently highly promoted by the IPCC (Doblas-Reyes et al., 2021). The definition of a "storyline" is given at the beginning of chapter 2.2.2, however, we agree that a short explanation in the introduction may improve the understanding of the terminology. In the revised manuscript, we added the definition of storylines.

  **Change 2:**

  Lines 33-36: Focusing on the rainfall event in 2009, Maraun et al. (2022) analysed the effect of projected future climate (2070–2100) and LULC changes on landslide occurrences  for the most-affected Feldbach region. In Maraun et al. (2022) the concept of storylines - simulations of physically self-consistent, plausible pathways of a specific event (Shepherd et al., 2018), was first applied in a landslide context, thus asking "how would this event occur in a warmer/colder climate, and what would be the associated landslide susceptibility".

  Lines 129-134: In an event storyline approach, the emphasis is placed on a qualitative understanding and plausibility of driving factors involved in an event, and thus the physically self-consistent unfolding of past, or plausible future events or pathways is examined (Shepherd et al., 2018). This implies that we are not investigating all future expected extreme rainfall events, but rather a similar event manifesting itself differently in varied climate scenarios. Recently, the Intergovernmental Panel On Climate Change (IPCC) emphasised the utility of the storyline approach for constructing and communicating regional climate information (Doblas-Reyes et al., 2021).

- **Point 3:** You defined the events that occurred in June 2009 and September 2014 as "extreme". How can you classify such events as "extreme"? Was a statistical analysis carried out?

  **Response 3:** The reviewer correctly pointed out a possible source of confusion. For the 2009 event, Haiden (2009, 4) analyzed the amount of rainfall in less than 24 h in Styria, and grouped that rainfall event into

events with a 50-year return period. Such an analysis was, however, not conducted for the 2014 event (nevertheless, as consequence, flood events corresponding to HQ50 and HQ100 were recorded at some places, see http://app.hydrographie.steiermark.at/berichte/september2014.pdf). Therefore, in the revised manuscript, we used the word "extreme" only in the context of the 2009 event, and "heavy" when referring to both events.

**Change 3:**

Lines 24-25: In June 2009 and September 2014, weather phenomena developed through a cut-off low brought heavy rainfall into the Styrian Basin, Austria (e.g., over 100 mm in 24 h in 2009).

Lines 27-28: The combined effect of premoisturing over the preceding winter and spring, and the occurrence of the actual triggering rainfall made these weather events into compound events [...].

Line 82: Study Area and  Rainfall Events

Line 91: In June 2009 and September 2014, heavy rainfall events occurred in southeast Styria.

Caption Figure 1: (b)  Landslide distribution for both rainfall events.

Line 120: As landslide data, we used landslides that occurred during the heavy rainfall events, which were initially mapped by [...].

– **Point 4:** You added in the susceptibility analysis the rainfall data on the landslide failure day, i.e. the triggering precipitation conditions. I think this is questionable and in contrast with the theoretical definition of susceptibility (see e.g. Reichenbach et al. (2018) [https://doi.org/10.1016/j.earscirev.2018.03.001]; van Westen et al. (2008) [https://doi.org/10.1016/j.enggeo.2008.03.010]). Landslide susceptibility is "the likelihood of a landslide occurring in an area on the basis of the local terrain and environmental conditions", therefore the triggering rainfall conditions should be removed from this analysis. You also wrote "For the landslide susceptibility analysis, we linked predisposing and triggering factors to landslide occurrences.". I think this can be considered a methodological issue.

**Response and Change 4:** *For the response and changes regarding the definition of landslide susceptibility, please refer to **Editor Point #1**.*

– **Point 5:** Regarding the environmental change simulation, you wrote (line 153) that "Adopting active forest management in the developed future LULC scenario, coniferous forest was replaced by climate resilient mixed forest".

**Response 5**: The authors agree with the referee that an in-depth analysis of effects of all possible kinds of land-use and land-cover changes on landslide occurrences is highly desirable. Actually, Maraun et al. (2022) analyzed a negative, "idealized" scenario representing extreme deforestation, i.e. one where all forest is removed (note: there was also a scenario with extreme afforestation). However, as the focus of the study is the assessment of uncertainties and not primarily the effect of LULC changes, we decided to include only the "realistic" instead of the "idealized" scenario ("realistic" scenario was developed in close cooperation with the Regional Forestry Directorate and the District Forestry

Authority). Furthermore, the LULC data used to fit our landslide model do not allow for further discrimination between different types of non-forested areas.

We added some explanations to refer the interested reader to Maraun et al. (2022) for these "idealized" scenarios.

**Change 5:**

Lines 166-167: The idealized LULC scenarios developed by Maraun et al. (2022) were not in the scope of this analysis (i.e. extreme de- and afforestation).

– **Point 6:** Furthermore, you wrote (line 161) "Specifically, for each grid cell we determined the maximum three-hour rainfall intensity, and we took the maximum five-day rainfall." In my opinion, also this is questionable, given that it is not always the most severe rainfall condition during a meteorological event that can trigger landslides. An explanation is needed.

**Response 6:** We agree with the reviewer that there may be rainfall-triggering landslide events for which other precipitation aggregation schemes are more appropriate. Generally speaking, there are many possibilities how to aggregate meteorological data, ranging from indices (e.g., landslide-rainfall index (Shou and Yang, 2015), antecedent rainfall index (Kirschbaum and Stanley, 2018)) to fixed moving sizes or the number of days exceeding a certain amount of rainfall (e.g., in Gassner et al. 2015; Kim et al. 2015), and there is no general agreement on an optimal aggregation scheme. However, in Knevels et al. (2020, 9), we compared the two meteorological variables to "rainfall events responsible for landslides" extracted using the approach of Melillo et al. (2015). Specifically, for our landslides, we discovered a correlation of 0.95 between five-day rainfall and total amount during rainfall event, and of 0.58 between maximum three-hour rainfall intensity and amounts of rainfall sub-events; confirming the applicability of the presented aggregation scheme. To inform the interested reader where the rainfall aggregation comes from, we have now added an explanation in an appropriate place.

**Change 6:**

Lines 111-114: Furthermore, precipitation was aggregated to obtain accumulated five-day rainfall (in mm) and maximum three-hour rainfall intensity (in mm h-1) on the landslide failure day, respectively (for a justification of the precipitation aggregation scheme, please refer to Knevels et al. 2020).

– **Point 7:** Finally, I suggest using round brackets for units of measurement. Please check all over the text.

**Response 7:** We are thankful for this hint. We replaced square-brackets with round brackets in the figures.

**Change 7:**

Figure 1: (mm); Figure 3: (mm), (mm h$^{-1}$), (%) + caption; Figure 4: (%); Figure 5a: (%); Figure A1: (mm); Figure A2: (%); Figure S1: (m); Figure S3: (mm), (mm h$^{-1}$), (%) + caption

**Anonymous Referee #2**

*This paper investigated the uncertainty cascade in storylines of landslide susceptibility emerging from climate change and parametric landslide model uncertainty. In general, this paper is interesting and rich in content. However, there are many mistakes in the basic concept of landslide susceptibility, hazard and risk assessment. In terms of landslide susceptibility prediction modeling process, the writing of this paper is rather rough. In terms of organizational structure, the thesis is difficult to understand. Therefore, it is recommended to reject the paper.*

- **Point 1:** Landslide susceptibility refers to the spatial probability of landslide occurrence affected by landslides themselves conditioning factors, without considering triggering factors such as heavy rainfall, earthquake, et al. Landslide hazard refers to the spatial and time probability of landslide occurrence under condition factors and trigger factors. Hence, this paper focus on landslide susceptibility affected by land cover change and heavy rainfall. I believe this paper has problems with the basic concepts of landslide susceptibility.

  **Response and Change 4:** *For the response and changes regarding the definition of landslide susceptibility, please refer to **Editor Point #1**.*

- **Point 2:** The writing ideas of this paper are very confused, and it is difficult for people to understand the specific steps and methods of the research. Especially in the introduction and methods section.

  **Response 2**: We are thankful for the referee pointing out sources of confusion and misunderstanding. Regarding the methods, we have added a flow chart in the appendix showing a general overview of methodical approach from the beginning - model fitting, to the end - uncertainty ratios. Regarding the introduction, we more specifically describe the term storyline and emphasize more clearly our objectives.

  **Change 2:**

  *For changes regarding the storyline approach, please refer to **Referee #1 Point #2**.*

  Lines 72-73: The focus was not on creating landslide susceptibility maps, but on the quantification of uncertainties.

  Lines 178-180: The proposed approach to assessing landslide susceptibility uncertainty addresses model fitting accounting for spatial dependency (Sect. 3.1), predictions considering environmental change (Sect. 3.2), and their amount of uncertainty (Sect. 3.3). Please refer to Fig. A1 in the Appendix A for an overview.

[Figure]

**Figure 1.** Overview of the proposed landslide susceptibility prediction and uncertainty assessment. (Note, the figure number does not match with the revised manuscript - Fig. A1 in the Appendix A)

– **Point 3:** Where are the input and output variables of landslide susceptibility prediction modeling described in this paper?

**Response 3:** The reviewer correctly pointed out a missing linkage to the Knevels et al. (2020) reference. The input variables were briefly mentioned at the second paragraph of the chapter in which the landslide model was introduced (3.1 Landslide Susceptibility Model, preprint lines 173-177). In Knevels et al. (2020) further details are available on how these input variables were derived (software, hyperparameter settings etc.). As this paper is already quite extensive in pages, we shortened this description. In the revised manuscript we added the information that more details on the delineation can be found in Knevels et al. (2020).

**Change 3:**

Lines 195-196: For more information on the delineation of the predictor variables and their maps, refer to Knevels et al. (2020) and Fig. S6 in the Supplementary Material, respectively.

– **Point 4:** How is the uncertainty problem concerned in this paper quantified?

**Response 4:** The authors thank the referee for the hint to a possible source of confusion. In last paragraph of the *chapter 3.3 Uncertainty Cascade in Landslide Predictions* (lines 285 - 289 in the preprint), the authors described how the ratios are derived: "Finally, to compare the different sources of uncertainty in the uncertainty cascade, we calculated the ratio of the uncertainty spread of within-event internal climate model variability and scenario uncertainty (i.e., spread of climate signals), respectively, to the parametric uncertainty spread of the landslide model. The joint uncertainty distribution of the landslide models' parametric uncertainty and within-event internal climate model variability is here referred to as the storyline uncertainty."

In the revised manuscript, we introduced a mathematical notation and changed the term "spread" to "width" to prevent the possibility of confusion. Furthermore, we hope that **Referee #2 Change #2** (flow chart of methodological approach) improves the understanding of the analysis.

**Change 4:**

Lines 304-311: (Note, the equation numbers do not match with the revised manuscript)

Finally, to compare the different sources of uncertainty in the uncertainty cascade, we calculated the following ratios:

$$R_{IV;Lsl} = \frac{width\ of\ CI_{IV}}{width\ of\ CI_{Lsl}} \tag{1}$$

$$R_{CS;Lsl} = \frac{width\ of\ CI_{CS}}{width\ of\ CI_{Lsl}} \tag{2}$$

where $R_{IV;Lsl}$ and $R_{CS;Lsl}$ denote the ratio of the uncertainty width of within-event internal climate model variability ($CI_{IV}$) and scenario uncertainty (i.e., width of climate signals, $CI_{CS}$), respectively, to the parametric uncertainty width of the landslide model ($CI_{Lsl}$). The joint uncertainty distribution of the landslide models' parametric uncertainty and within-event internal climate model variability is here referred to as the storyline uncertainty ($CI_{Story}$).

– **Point 5:** Where are the environmental factor maps of landslide susceptibility and landslide susceptibility outcome maps? These problems [Points 3-5] are not well reflected.

**Response 5**: The reviewer pointed out a possible source of misunderstanding. This article is not about creating landslide maps, but to quantify uncertainties in predictions emerging from the landslide and climate models. However, we followed the reviewer and added two overview maps in the supplementary materials: a map showing all the input variables, and a map showing the landslide susceptibility classes of the event and a selected climate model alongside with uncertainty graphs, which may be useful in communicating with local stakeholders.

Please note that the estimated predictor-response relationship, which were of interest in our analysis, were already made available via the supplementary materials (Figure S4 and S5, https://nhess.copernicus.org/preprints/nhess-2022-154/nhess-2022-154-supplement.pdf).

**Change 5:**

Lines 544-547: In the spatial context, recommendations for the construction of new settlement infrastructure may be derived based on a map, and zones of high landslide susceptibility along with uncertainty graphs (Fig. S7 in the Supplementary Material) can be communicated to local planners and environmental managers (cf. Petschko et al., 2014).

[Figure]

**Figure 2.** Overview of input variables for predicting landslide susceptibility of the 2009 event. Geology: 0: 'Others', 1: 'Neogene formations dominated by fine-grained sediments', 2: 'Neogene formations with coarse-grained layers', 3: 'pre-Würmian Pleistocene formations', 4: 'Würm and Holocene sediments'. (Note, the figure number does not match with the revised manuscript - Fig. S6 in the Supplementary Material)

[Figure]

**Figure 3.** Example of landslide susceptibility maps of the 2009 event and associated uncertainties in highly susceptible areas. Pre-industrial and future storylines are based on the HadGEM climate model in the 4 K warming scenario. Note: Uncertainty graphs are based on the entire study area. (Note, the figure number does not match with the revised manuscript - Fig. S7 in the Supplementary Material)

– **Point 6:** The figures are not clear enough.

**Response 6:** The authors collectively went through all figures individually, and agreed that, except for Figure 5, all figures are reasonably clear in spite of their high information content. The authors modified Figure 5 by adding the mathematical notations introduced with **Referee #2 Change #4**, and thus improved its understanding.

**Change 6:** see Figure 5 in the revised manuscript.

– **Point 7:** The references are not new enough and are not representative enough.

**Response 7:** The authors checked all references again on redundancy or the possibility for "updating", and modified them when it suited.

**Change 7:**

Removed: Hastie and Tibshirani 1986; Nychka 1988; Oberkampf et al. 2004; Refsgaard et al. 2007; Brenning 2012; Fressard et al. 2014; Hussin et al. 2016

Added: Brock et al. 2020; Ozturk et al. 2021; Jones et al. 2021

– **Point 8:** The uncertainty characteristics are assessed by which indexes? These description are not clear.

**Response and Change 8:** Please refer to **Referee #2 Point #4**.

– **Point 9:** There is insufficient analysis of feasible solutions to the problems in this paper.

**Response 9:** The problem addressed in this paper is the assessment of uncertainties - not the reduction of uncertainties. Nevertheless, the authors even gave some ideas, how these analyzed uncertainties might be reduced (in the preprint: e.g. Lines 440, 450-453 for the climate uncertainties or Lines 482-485, 515-516 for the landslide model uncertainties). The authors have shown that the chosen approach to uncertainty assessment is valid and (evidently) feasible. We are therefore confident that in reviewing these text sections, the referee will appreciate our contributions mainly to the assessment of uncertainties.

**Change 9:** No change made